# A comprehensive transformer-based approach for high-accuracy gas adsorption predictions in metal-organic frameworks

Jingqi Wang[1,2,7], Jiapeng Liu [3,4,7], Hongshuai Wang [2,5], Musen Zhou [6], Guolin Ke [2], Linfeng Zhang[2,4], Jianzhong Wu [6] ✉, Zhifeng Gao [2] ✉ & Diannan Lu [1] ✉

Gas separation is crucial for industrial production and environmental protection, with metal-organic frameworks (MOFs) offering a promising solution due to their tunable structural properties and chemical compositions. Traditional simulation approaches, such as molecular dynamics, are complex and computationally demanding. Although feature engineering-based machine learning methods perform better, they are susceptible to overfitting because of limited labeled data. Furthermore, these methods are typically designed for single tasks, such as predicting gas adsorption capacity under specific conditions, which restricts the utilization of comprehensive datasets including all adsorption capacities. To address these challenges, we propose Uni-MOF, an innovative framework for large-scale, three-dimensional MOF representation learning, designed for multi-purpose gas prediction. Specifically, Uni-MOF serves as a versatile gas adsorption estimator for MOF materials, employing pure three-dimensional representations learned from over 631,000 collected MOF and COF structures. Our experimental results show that Uni-MOF can automatically extract structural representations and predict adsorption capacities under various operating conditions using a single model. For simulated data, Uni-MOF exhibits remarkably high predictive accuracy across all datasets. Additionally, the values predicted by Uni-MOF correspond with the outcomes of adsorption experiments. Furthermore, Uni-MOF demonstrates considerable potential for broad applicability in predicting a wide array of other properties.

Gas separation[1–3] is a significant industrial challenge that requires immediate attention, given its critical role in various applications. For example, separating $CH_4$ from $CO_2$ is essential for obtaining high-quality natural gas and effectively achieving the carbon capture, utilization and storage for environmental reasons[4,5]. Gas separation also has implications for other fields, such as the production of high-purity oxygen[6,7] and nitrogen[8,9] for industrial purposes and the purification of noble gases for medical diagnosis, and lasers[10,11]. Given its

[1]Department of Chemical Engineering, Tsinghua University, Beijing 100084, China. [2]DP Technology, Beijing 100089, China. [3]School of Advanced Energy, Sun Yat-Sen University, Shenzhen 518107, China. [4]AI for Science Institute, Beijing 100190, China. [5]Jiangsu Key Laboratory for Carbon-Based Functional & Materials Devices, Institute of Functional & Nano Soft Materials (FUNSOM), Soochow University, Suzhou 215123, China. [6]Department of Chemical and Environmental Engineering, University of California, Riverside, CA 92521, USA. [7]These authors contributed equally: Jingqi Wang, Jiapeng Liu. ✉e-mail: jwu@engr.ucr.edu; gaozf@dp.tech; ludiannan@mail.tsinghua.edu.cn

importance, research in this area is crucial for advancing technology and meeting industry demands.

Metal-organic frameworks (MOFs) have emerged as a kind of promising material in the field of gas separation due to their unique properties[12–14]. MOFs are composed of metal ions and organic ligands, which provide them with highly ordered pore structures and adjustable aperture sizes. These properties make them ideal for various gas separation applications[15–17]. The ability to control the pore size and chemical composition of MOFs allows for selective adsorption and separation of different gases. MOFs with different pore sizes exhibit varied capacities for gas adsorption[18], and tuning the chemical composition[19] can affect the preference for adsorbate gases. The ability to selectively adsorb and separate different gases makes MOFs a promising material for various industrial and environmental applications[20,21].

While the potential of MOFs for gas adsorption is promising, accurately predicting their adsorption capacity remains a challenge. Molecular dynamics[22,23], Monte Carlo (MC)[24] and other simulation/calculation methods[25,26] have been applied to provide reference values, but these approaches are computationally expensive and complicated for implementation, limiting their application to large-scale, multi-gas, and high-throughput calculations. Moreover, the vast range of operating conditions for gas adsorption further complicates the predictions.

Machine learning techniques have demonstrated significant potential in accurately predicting properties of crystalline materials[27–29], reducing the cost of traditional trial and error experiments, and eliminating the need for expensive simulations. However, these methods often rely on feature engineering based on expert domain knowledge, leading to overfitting and biased performance when using a limited amount of labeled data. With the emergence of deep learning, graph neural networks,[30,31] and Transformers[32–34] have proven successfully in predicting MOF properties. These models directly incorporate structural information such as chemical bonds, atoms, and spatial coordinates as inputs, and automatically learn structural features through data-driven approaches. Thanks to the powerful representation capacity of these deep-learning frameworks, learned features can effectively eliminate biases introduced by feature engineering mentioned earlier.

Despite their high performance and powerful predictive capabilities, existing models for predicting adsorption properties are typically designed for single tasks, specifically predicting the adsorption uptake of a particular gas under certain conditions. However, the available datasets for these single task predictions are often limited, thereby hindering the models generalizability and full utilization of their capabilities. On the other hand, the combination of labeled data from various adsorbate gases across different temperature and pressure environments can create a substantial dataset suitable for training across the entire working conditions. The increased data size may also enhance the model ability to generalize and improve their practical industrial use. Therefore, a unified adsorption framework is necessary for advancing these models. Additionally, integrating representation learning (or pre-training) for large-scale unlabeled MOF structures may further improve the model performance as well as representation ability. The pre-training trick has been widely implemented in combination with large-scale models to discover drugs[35], where pure three-dimensional molecular structures were used to pre-train these models. Experimental results have also demonstrated that pre-trained models outperform previous methods, particularly in property prediction[33], suggesting remarkable improvement through pre-training.

Inspired by this, we propose the Uni-MOF framework as a multi-purpose solution for predicting gas adsorption of MOFs under different conditions using structural representation learning. Compared with other Transformer-based models such as MOFormer[34] and MOFTransformer[33], our Uni-MOF, as a Transformer-based framework,

not only can the pre-training recognize and recover the three-dimensional structure of nanoporous materials and thus greatly improve the robustness of the model, but the fine-tuning task also further takes into account the operating conditions such as temperature, pressure, and different gas molecules, which makes Uni-MOF suitable for both scientific research and practical applications. Uni-MOF, as a comprehensive gas adsorption estimator for MOF materials, requires only the crystallographic information file (CIF) of the MOF, along with the associated gas, temperature, and pressure parameters, to predict the gas adsorption properties of nanoporous materials over a wide range of operating conditions. Our framework is easy to use and allows for module selection. Additionally, it effectively addresses the issue of overfitting by integrating various cross-system absorption labeled data with representation learning from massive amounts of unlabeled structural data. This compensates for the lack of high-quality and insufficient data, ultimately leading to higher accuracy in gas adsorption predictions. Our study utilized a self supervised learning approach on a database containing over 631,000 MOF and COF structures. The results were remarkable, demonstrating a high prediction accuracy. Fine-tuning experiments revealed that the Uni-MOF framework is robust in databases with ample data. When applied to databases with sufficient sampling of working conditions, our Uni-MOF framework is able to screen high performance adsorbents under high pressure accurately by feeding only the labeled data obtained at low pressure by home-brew simulations. We must stress that Uni-MOF provides a convenient approach for high pressure adsorption capacities, which are generally more computationally demanding for traditional simulations. The results are consistent with experimental screening outcomes. Furthermore, the performance of Uni-MOF on cross-system datasets exceeded that on single-system tasks. By leveraging support from other gas adsorption data, Uni-MOF accurately predicted the adsorption properties of unknown gases.

Uni-MOF framework achieves material recognition accuracy at the atomic level, while the integrated model makes Uni-MOF more applicable to engineering problems. Undoubtedly, accomplishing truly unified models is the future direction of the materials field, rather than just focusing on specialized areas. Uni-MOF is a pioneering practice of Machine Learning in gas adsorption.

## Results and discussion
### Overview of workflow

The Uni-MOF framework comprises pre-training on three-dimensional nanoporous crystals and fine-tuning for multitask prediction in downstream applications. Figure 1 provides a schematic representation of the Uni-MOF framework. The pre-training of three-dimensional crystal materials significantly enhances the prediction performance of downstream tasks, particularly for large-scale unlabeled data. To address the issue of inadequately supervised training datasets, we collected an extensive dataset of MOF structures and generated over 300,000 MOFs using ToBaCCo.3.0[36,37]. High-throughput construction of COFs based on materials genomics strategy with quasi-reactive assembly algorithms (QReaxAA) is feasible, leading to a comprehensive library of COFs[38]. Through the spatial configuration of materials, Uni-MOF is capable of learning the material structural properties, most importantly the chemical bonding information, very well. In order to enable Uni-MOF to learn more diverse materials and thus improve the generalization ability to a broader range of materials, we introduced MOFs and COFs both virtually and experimentally during the pre-training process. Similar to the masking tagging task in BERT[39] and Uni-Mol[35], Uni-MOF employs a prediction task for masked atoms, thereby promoting the pre-trained models to acquire an in-depth understanding of the material spatial structures. To enhance the robustness of pre-training and generalize the learned representation, we introduced noises to the original coordinates of MOFs, as depicted in Fig. 1a. In the pre-training stage, we devised two tasks. 1) reconstructing the

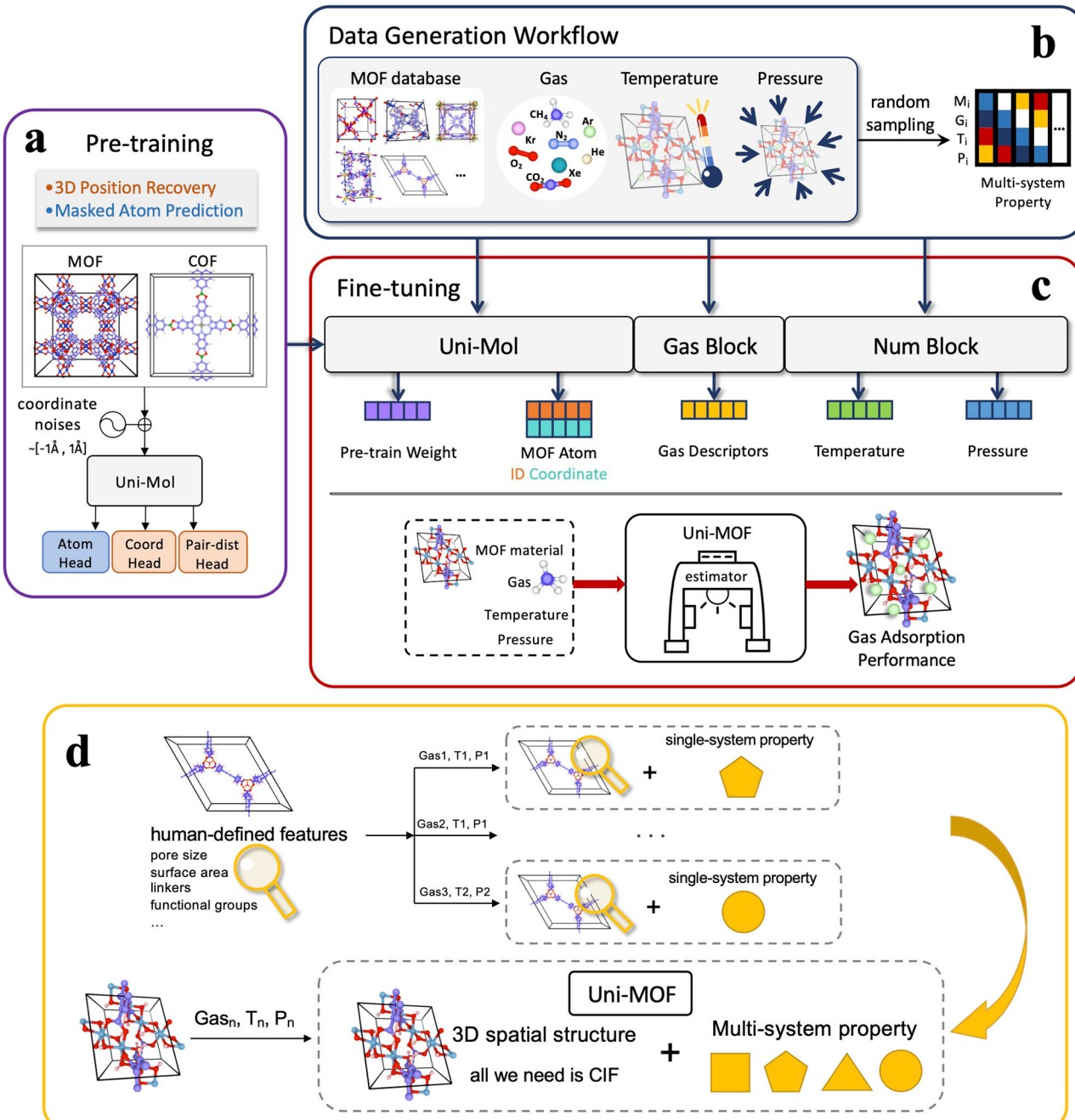

**Fig. 1 | Schematic overview of Uni-MOF framework. a** pre-training workflow. In the pre-training stage, in addition to predicting the types of masked atoms, a three-dimensional position denoising task was used to learn the three-dimensional spatial representation. Uniform noise of [−1 Å, 1 Å] is added to the 15% of atomic coordinates randomly, and then the spatial position encoding is calculated based on the corrupted coordinates. **b** data generation workflow. Cross-system performance datasets can be collected or generated by random sampling of different operating conditions. **c** workflow of Uni-MOF fine-tuning. A unified gas adsorption prediction model Uni-MOF is built by the embedding of pre-trained weight, MOF material, gas, temperature, and pressure. **d** overall workflow of Uni-MOF. For the multi-purpose Uni-MOF framework, no additional analytical calculations for materials are required, and the properties under varied working conditions can be predicted based solely on the crystallographic information file (CIF) of MOF materials. MOF means metal-organic framework.

pristine three-dimensional positions from the noisy data, and 2) predicting the masked atoms. These tasks can augment the model robustness and improve downstream prediction performance.

In addition to diverse spatial configurations, a comprehensive set of material property data points is also crucial for model training. To enrich the dataset, we established a custom data generation process (as illustrated in Fig. 1b). For example, we utilized the CoRE MOF database that comprises successfully synthesized MOFs, gases that are significant in the separation field, as well as the common temperature

and pressure operating range under the corresponding system. By randomly sampling from these various materials, gases, temperatures, and pressure pools, a significant volume of adsorption uptake data can be generated for Uni-MOF fine-tuning. This data generation process improves the efficiency of data generation and can form a widely sampled dataset, Table 1 lists all the databases applied in this study. The simulation-derived database with diversity is beneficial for model fine-tuning and optimization, ultimately accomplishing the objective of virtual screening for material performance.

**Table 1 | Structure & Data resources**

| Data Types | Sources | Availability | Software | Size |
|---|---|---|---|---|
| Structure | collection | hMOF[50] | — | 137,000+ |
| Structure | collection | ToBaCCo | — | 10,000+ |
| Structure | collection | CoRE MOF[51] | — | 12,000+ |
| Structure | collection | CCDC | — | 12,000+ |
| Structure | collection | CoRE COF[52] | — | 600+ |
| Structure | collection | GCOFs[38] | — | 160,000+ |
| Structure | generation | this work | ToBaCCo.3.0 | 300,000+ |
| Adsorption Uptake Data | collection | hMOF_MOFX_DB | — | 2,400,000+ |
| Adsorption Uptake Data | collection | CoRE_MOFX_DB | — | 460,000+ |
| Adsorption Uptake Data | generation | this work | RASPA[54] | 99,000+ |
| Other Property Data | generation | this work | Zeo++[60] | 149,000+ |

The fine-tuning of Uni-MOF depicted in Fig. 1c is based on the extraction of representations acquired through pre-training, as well as the generation and collection of extensive datasets using our home-brew workflows. During the fine-tuning process, we trained the model using around 3,000,000 labeled data points across various adsorption conditions of MOFs and COFs, enabling accurate prediction of adsorption capacities. With the diverse database of cross-system targeted data, the fine-tuned Uni-MOF can predict the multi-system adsorption property of MOFs under arbitrary states, including different gases, temperatures, and pressures. As a result, Uni-MOF is a unified and readily available framework for predicting the adsorption performance of MOF adsorbents.

Above all, Uni-MOF obviates the requirement for additional labor in identifying human-defined structural features. Instead, the CIF of MOFs, along with pertinent gas, temperature, and pressure parameters, suffices. The self-supervised learning strategy and abundant databases ensure that Uni-MOF can foretell gas adsorption properties of nanoporous material in a wide range of operating parameters, thereby rendering it a proficient gas adsorption estimator for MOF materials.

## Overall performance in large-scale databases

In order to evaluate the predictive capability of Uni-MOF as a comprehensive framework for adsorption performance prediction, two mixed-state databases for gas adsorption, namely hMOF_MOFX-DB and CoRE_MOFX-DB, were compiled with adsorbate gases consisting of [$CO_2$, $N_2$, $CH_4$, $Kr$, $Xe$] and [$N_2$, $Ar$], respectively. In addition, the CoRE_MAP_DB database was generated via our home-brew MC simulation workflow for the adsorption uptake of seven gases ($CO_2$, $CH_4$, $Ar$, $Kr$, $Xe$, $O_2$, $He$).

Since the data sources of the three databases are different, we conducted model training for each database separately in order to ensure the consistency of data sets. Details of these three databases, including temperature and pressure ranges, are listed in Supplementary Table 1. To prevent data bias and ensure that the test set remained unseen by the model, we divided the data set into three different data sets (train, valid, and test) with the ratio of 8:1:1 according to the MOF structure instead of randomly splitting, that is, there is no identical material between the three datasets. The splitting ensures that the model accomplishes the prediction of new materials in validation and test set, rather than those materials that have already been seen. The model parameters are optimized during the training process and reflected in the results of the validation set. The optimal model corresponding to the validation set is saved, and the prediction results in the never-before-seen test set represent the final performance ($R^2$ shown here) of the model, thus reasonably avoiding over-fitting.

The collected datasets, hMOF_MOFX_DB and CoRE_MOFX_DB, exhibit relatively concentrated temperature and pressure distributions, as depicted in the sub-figure of Fig. 2a, b. Notably, both databases offer adequate data, with over 2,000,000 and 400,000 data points, respectively. The prediction results demonstrate that Uni-MOF is remarkably robust when applied to databases that possess sufficient data with relatively concentrated operating states, such as hMOF_MOFX_DB and CoRE_MOFX_DB, with $R^2$ values of 0.98 and 0.92, respectively. In contrast, our CoRE_MAP_DB database, which we have learned from Supplementary Table 1, contains slightly less than 100,000 data points. It encompasses an extensive sampling of the adsorption of seven adsorbed gases in over 10,000 MOFs, covering a temperature range of 150–300 K and a pressure range of 1 Pa–3 bar, as depicted in the sub-figure of Fig. 2c. For such a widely distributed database, Uni-MOF can still achieve excellent prediction accuracy with an $R^2$ value of 0.83, demonstrating its good generalizability.

The analysis also incorporates two other error metrics, i.e., Mean Absolute Error (*MAE*) and Root Mean Square Error (*RMSE*). However, the CoRE_MOFX dataset shows larger errors, particularly in *RMSE*. CoRE_MOFX dataset contains Ar and $N_2$ adsorption amounts at 77 K and 87 K, which are significantly lower temperatures compared to the other two databases. Consequently, the adsorption values in CoRE_MOFX are generally larger. The adsorption isotherms with the most significant errors are depicted in Supplementary Fig. 1. MOF materials with the largest error typically have Cu (with the lowest energy parameter of 2.52 K) as the metal site. This implies that MOF materials with low energy parameter metal sites or large pore sizes are the limiting factors for low-temperature data prediction. Intriguingly, even with the highest errors, Uni-MOF can accurately reproduce the adsorption trend from low to high pressure. Additionally, CoRE_MOFX_DB, as the collected experimental structural database, contains 1800+ disordered materials out of 12,000+ MOFs. Disordered MOFs were found to adversely affect the model performance, as the unrealistic internal structures and limited material samples lead to the challenging prediction of gas adsorption, as demonstrated by the outlier ja5111317_ja5111317_si_003_clean and LELDOX_clean in Supplementary Table 3. Therefore, disordered materials were excluded from the generated CoRE_MAP_DB database, which results in reduced errors and is more suggested to use for prediction of gas adsorption in nanoporous materials. The force field used in this work does not account the effect of open-metal sites. For the top ten outliers in the CoRE_MOFX_DB database (shown in Supplementary Table 4), 80% of them have open-metal sites. This suggests that the significant deviations between simulations and Uni-MOF predictions could be due to the missing interaction between open-metal sites and adsorbate considered in the simulation.

Additionally, we discovered that the prediction accuracy in the pre-training stage could reach 0.98 before and after incorporating COF materials. This indicates that Uni-MOF is capable of effectively learning the three-dimensional spatial structure of multiple nanoporous materials. As for the downstream tasks, the predictive performance ($R^2$, *RMSE*, and *MAE*) of Uni-MOF for all three databases surpassed that of Uni-MOF with only MOF pre-training, as illustrated in Supplementary Tables 5, 6. This indicates that incorporating COF not only maintains but also enhances the model robustness, further demonstrating the superiority of our Uni-MOF framework.

## Experimental adsorption uptake prediction

Despite the excellent prediction performance of Uni-MOF on simulated results, we are wondering how the framework would behave in comparison to the real experimentally collected results. In this study, we have chosen commonly used laboratory materials, such as MOF-5 and MOF-177[40–43], and compared the predicted gas adsorption capacity with existing experimental data. Considering that the CoRE_MAP_DB database contains a diverse range of operating conditions and

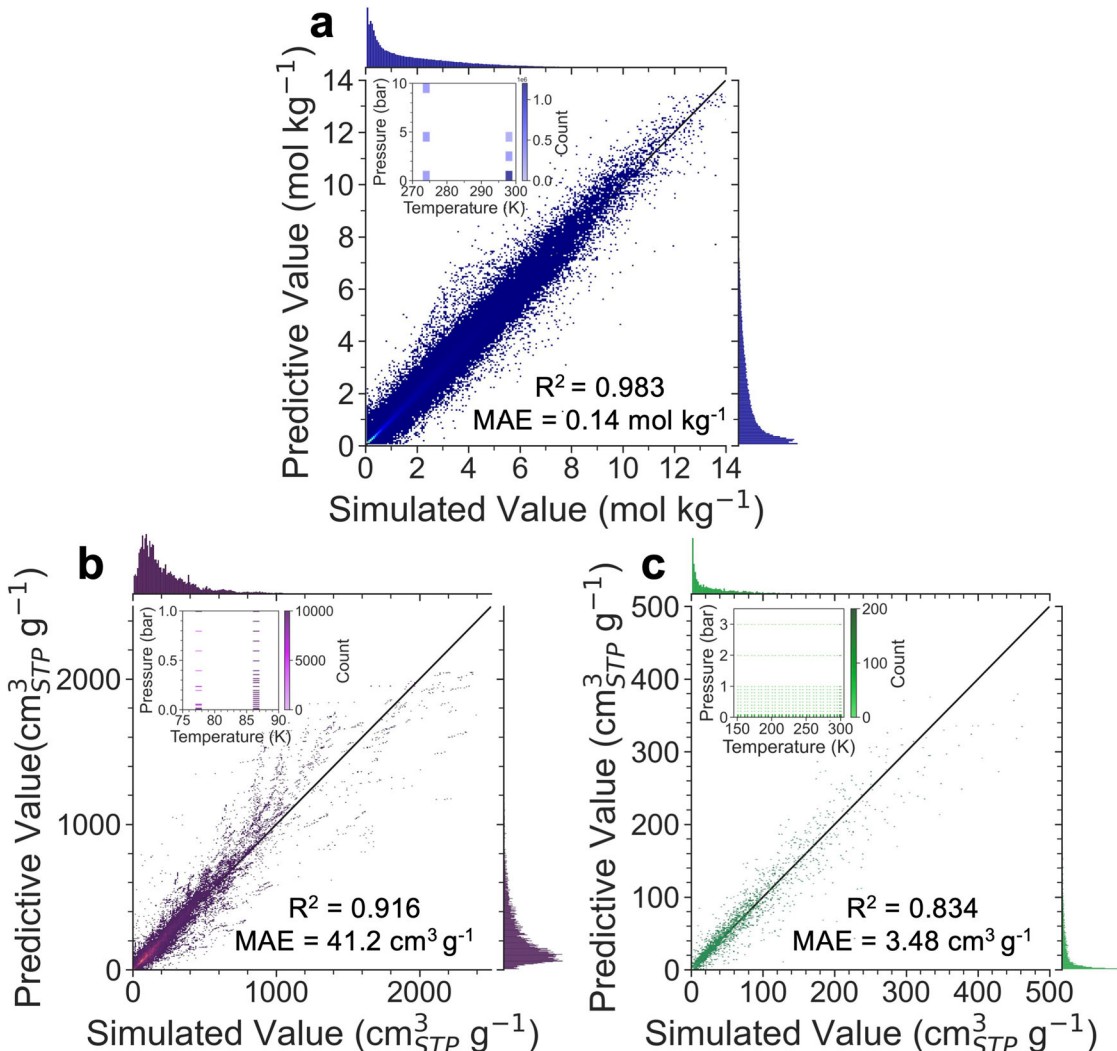

**Fig. 2 | Overall performance of Uni-MOF in large-scale databases.** The correlation between predicted and simulated value of gas adsorption amount for (**a**) Database of hMOF_MOFX_DB (mol·kg$^{-1}$), (**b**) Database of CoRE_MOFX_DB (cm$^3_{STP}$ · g$^{-1}$) and (**c**) Database of CoRE_MAP_DB (cm$^3_{STP}$ · g$^{-1}$). Sub-figure is the distribution of temperature (K) and pressure (bar) for each database. $R^2$ means coefficient of determination, MAE represents mean absolute error, MOF means metal-organic framework. Source data are provided as a Source Data file.

experimentally validated MOF structures, we chose the weights trained using this database to predict the adsorption performance of experimental materials. Figure 3 displays the predicted results under different conditions, details are listed in Supplementary Tables 8–10.

Figure 3a presents the Langmuir adsorption isotherm[44] obtained by fitting the predicted methane adsorption capacity under low pressure (less than 5 bar). It shows that the high pressure adsorption capacity displayed by the Langmuir adsorption isotherm is consistent with the experimental values. Specifically, the high pressure adsorption capacity of (Zn2(bdc)2(dabco), methane, 393 K) > (MIL-101, methane, 373 K) > (MOF-177, methane, 398 K). This suggests that the Uni-MOF framework is capable of accurately screening high performance adsorbents under high pressure based solely on prediction adsorption capacity under low pressure. Nevertheless, we still notice significant deviations between many predicted and experimental values under low pressure, especially in the cases of Mg-dobdc and MOF-5. Simulations may not precisely represent experimental values for MOFs with open metal sites, such as Mg-dobdc[45]. However, MOF-5 has close metal sites, and previous studies have shown that simulations can effectively depict experimental gas adsorption values[46]. Despite these findings, we observed that the gas adsorption performance of MOFs varies even under the same operating conditions (refer to

Supplementary Tables 8–10). This suggests that different methods and significant objective errors exist in the experimental values. Therefore, we did not purposely select data but instead aimed to provide as comprehensive a representation of the collected data as possible. For example, the black intersections in Fig. 3e represent experimental values from various literature sources under identical operating conditions. The results demonstrate that, despite significant variations in experimental values, the Uni-MOF framework maintains a high level of accuracy in material ranking, rendering it suitable for addressing engineering challenges.

Furthermore, the experimental values are introduced to correct the adsorption isotherms. For example, Fig. 3b shows the Langmuir adsorption isotherm obtained by fitting both the predicted and experimental adsorption data. While we use simulated datasets to address data scarcity, we can also properly introduce experimental values to correct adsorption isotherms, which helps a more quantitative prediction of adsorption performance at high-pressure where the gas-gas interaction becomes more significant. In Fig. 3b, one can observe that the corrected adsorption isotherms have a strong correlation with experimental adsorption capacity to some extent. The results exhibit that Uni-MOF not only has the ability to screen the adsorption performance of the same gas in different materials but also

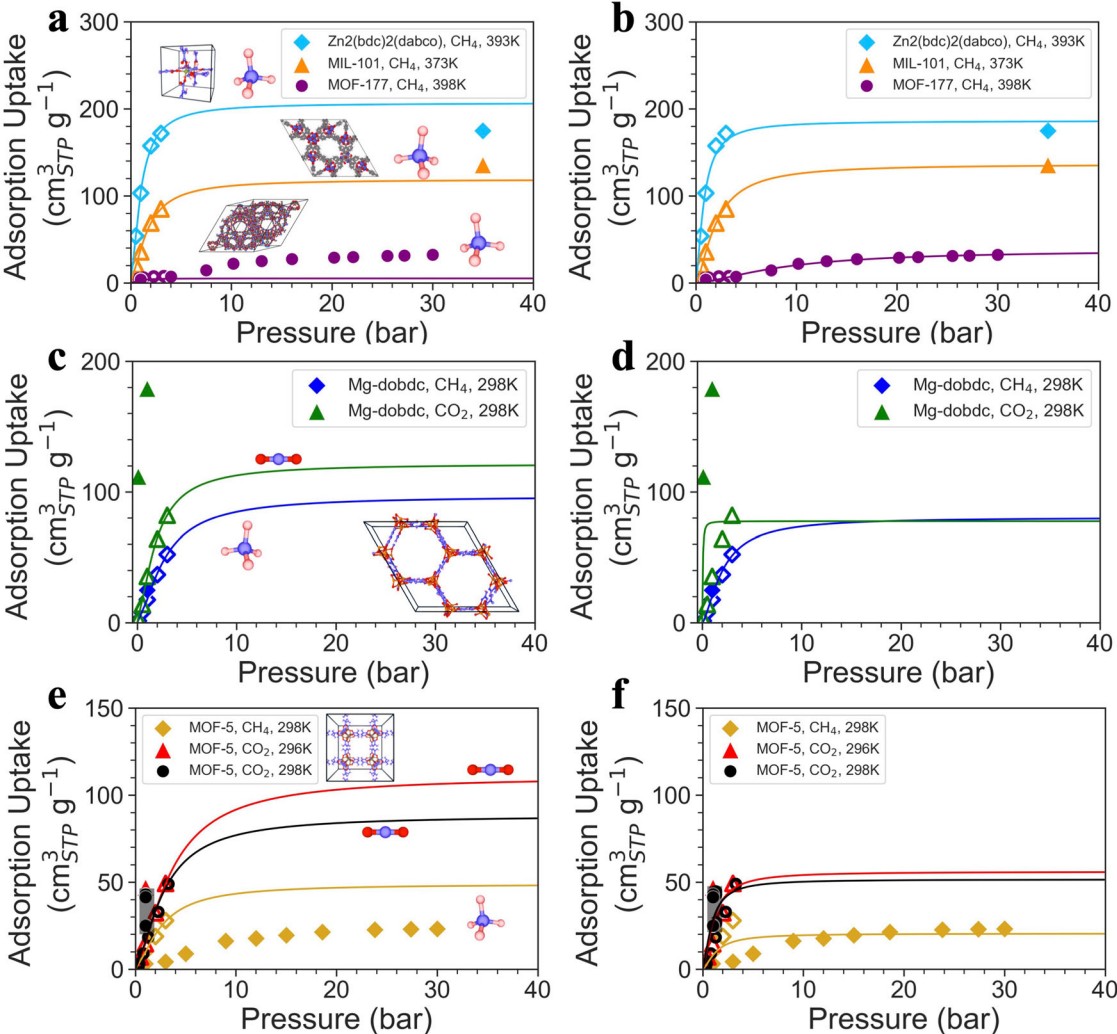

**Fig. 3 | Adsorption isotherms based on low-pressure predictions and high-pressure experimental values, each curve represents Langmuir fit. a** Uni-MOF predicted and (**b**) experimentally corrected adsorption isotherms of methane adsorption in Zn2(bdc)2(dabco), MIL-101 and MOF-177 at 393 K, 373 K, and 398 K, respectively. **c** Uni-MOF predicted and (**d**) experimentally corrected isotherms of methane and carbon-dioxide adsorption in Mg-dobdc at 298 K. **e** Uni-MOF predicted and (**f**) experimentally corrected adsorption isotherms of methane and carbon-dioxide adsorption in MOF-5 at 298 K and 296 K. Each set of horizontal plots (i.e., **a** and **b**, **c** and **d**, **e** and **f**) represents the same system, that is, the same MOF and gas molecule. The hollow marker represents the predicted value of Uni-MOF, and the filled marker represents the experimental data referenced from previous literature.

can accurately screen the adsorption performance of different gases in the same material (Fig. 3c, d) or at different temperatures (Fig. 3e, f).

In the foreseeable future, the intersection of Artificial Intelligence (AI) and materials science will necessitate the resolution of practical and scientific issues. Nonetheless, the attainment of process implementation by AI in the realm of machine learning techniques that entail copious amounts of data remains a formidable challenge, given the dearth of experimental data and the diverse array of synthetic technology and characterization conditions implicated. Our research has made a significant stride in materials science by incorporating operating conditions into the Uni-MOF framework to ensure data adequacy and enable screening functions that are consistent with experimental findings.

**Cross-system forecasting**

In order to showcase the predictive capabilities of Uni-MOF with regard to cross-system properties, five materials were randomly selected from each of the six systems (carbon-dioxide at 298 K, methane at 298 K, krypton at 273 K, xenon at 273 K, nitrogen at 77 K and argon at 87 K) contained in databases hMOF_MOFX_DB and

CoRE_MOFX_DB, which have been thoroughly sampled in terms of temperature and pressure. The predicted and simulated values of gas adsorption uptake at varying pressures were then compared, with the results presented in Fig. 4a–f. Adsorption isotherms fitting from both Uni-MOF predictions and simulated values would artificially reduce visual errors. In order to eliminate data bias, adsorption isotherms in all cases were obtained only by simulated values. It is evident that, due to the fact that the adsorption isotherms were obtained purely through simulated values, the predicted values of adsorption uptake generated by Uni-MOF for the hMOF_MOFX_DB and CoRE_MOFX_DB databases align closely with the simulated values across all cases. This finding is further supported by the high prediction accuracy demonstrated in Fig. 2a, b.

Given the ability of Uni-MOF to predict properties across systems, we were intrigued by its potential to forecast the adsorption capacity of unknown gases. The CoRE_MAP_DB database contains a diverse array of adsorption data points for various gases, such as methane, carbon dioxide, argon, krypton, xenon, oxygen, and helium. To evaluate the predictive capability of Uni-MOF, we randomly divided the CoRE_MAP_DB data points by adsorbate gas into three datasets (train,

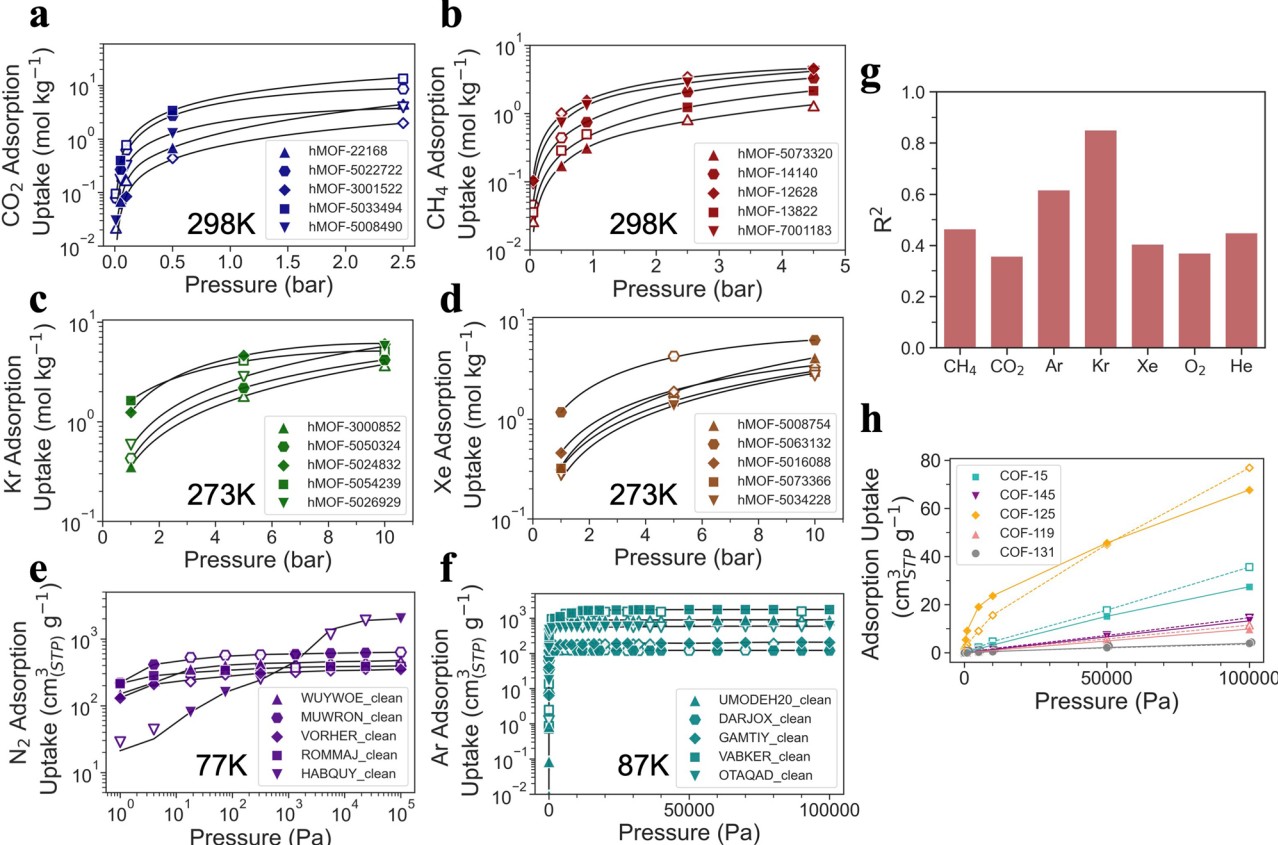

**Fig. 4 | Uni-MOF cross-system prediction cases.** The predicted and simulated values of gas adsorption amount versus pressure in hMOF_MOFX_DB database for (**a**) Carbon-dioxide at 298 K, (**b**) Methane at 298 K, (**c**) Krypton at 273 K and (**d**) Xenon at 273 K, and in CoRE_MOFX_DB database for (**e**) Nitrogen at 77 K and (**f**) Argon at 87 K. Different markers in each figure denote different MOF adsorbents. The hollow marker represents the predicted adsorption uptake of Uni-MOF, and the filled marker represents the simulated adsorption uptake in the database. The adsorption isotherms are obtained only by simulated values to eliminate data bias.

**g** Gas transfer learning under multi-system conditions, with the random division into three datasets (train, validation, and test dataset with a ratio of 5:1:1) according to adsorbate gases. **h** Predicted and simulated $CH_4$ adsorption isotherms at 300 K in COF materials, different marker types represent different COFs. The hollow marker means predicted value, the filled marker represents simulated value. $R^2$ represents the coefficient of determination. Source data are provided as a Source Data file.

valid, and test dataset with a ratio of 5:1:1), then predicted the adsorption capacity for each gas separately. The resulting predictions are depicted in Fig. 4g and summarized in Supplementary Table 11. Remarkably, Uni-MOF demonstrated robustness in predicting the adsorption capacity of unknown gases, achieving a high prediction accuracy ($R^2$) of 0.85 for krypton and a prediction accuracy above 0.35 for all unknown gases.

One can observe that prediction performance varies among gases. For instance, Ar, Kr, and Xe are all noble gases with Kr demonstrating the best performance among them with $R^2$ of 0.85, while the prediction performance of Xe is inferior ($R^2 = 0.41$). A similar distribution of MOF material features across Ar, Kr, and Xe databases is observed, as illustrated in the corresponding sub-figure of Supplementary Fig. 2, indicating that MOF material sampling differences have minimal impact in this case. Despite the larger molecular diameter of Xe, making it less favorable for adsorption in small pores, it shows a higher adsorption limit (depicted in Supplementary Fig. 3) compared to argon and krypton. This may be a result of the greater energy parameters of Xe ($\epsilon = 167.1$ K) when interacting with MOF material atoms. In the sub-figure of Supplementary Fig. 2c, materials with high Xe adsorption capacity (exceeding 400 $cm^3 g^{-1}$) are highlighted in yellow. These materials generally exhibit larger pore sizes and void fractions, possessing an largest cavity diameter (LCD) greater than 6.73 Å, a void fraction exceeding 0.68, and a PLD of at least 4.18 Å (close to the kinetic diameter of xenon with 3.96 Å). Although it is challenging to

establish a comprehensive connection between model predictions and the physicochemical properties of gases, it is discernible that gases possessing larger energy parameters typically demonstrate inferior predictive performance, such as $CH_4$, $CO_2$, and Xe. This phenomenon can be attributed to the increased significance of gas molecule interactions with escalating pressure, coupled with the complexity introduced by high energy parameters, ultimately complicating transfer learning.

Notwithstanding the challenges, encouraging outcomes can still be observed, as the model precisely predicts the adsorption of moderate krypton, premised on the adsorption behavior of argon and xenon within the same inert gas category.

In this study, we showcase the general performance of Uni-MOF for gas transfer learning, employing a database comprising fewer than 100,000 data points. This suggests that even the optimal prediction conditions were not specifically selected, Uni-MOF is capable in engineering domains. To mitigate over-fitting, the data is typically divided into three sets. However, transfer learning across diverse gases presents a considerable challenge, especially when confronted with a restricted number of gas varieties and data points. As a result, in order to optimize the utilization of the scarce data, the results obtained by dividing the data into two datasets based on the gas types are also shown in Supplementary Fig. 5. Results revealed an improvement in transfer learning accuracy for each gas to some extent. Unlike the prediction of adsorption between different materials, the prediction of

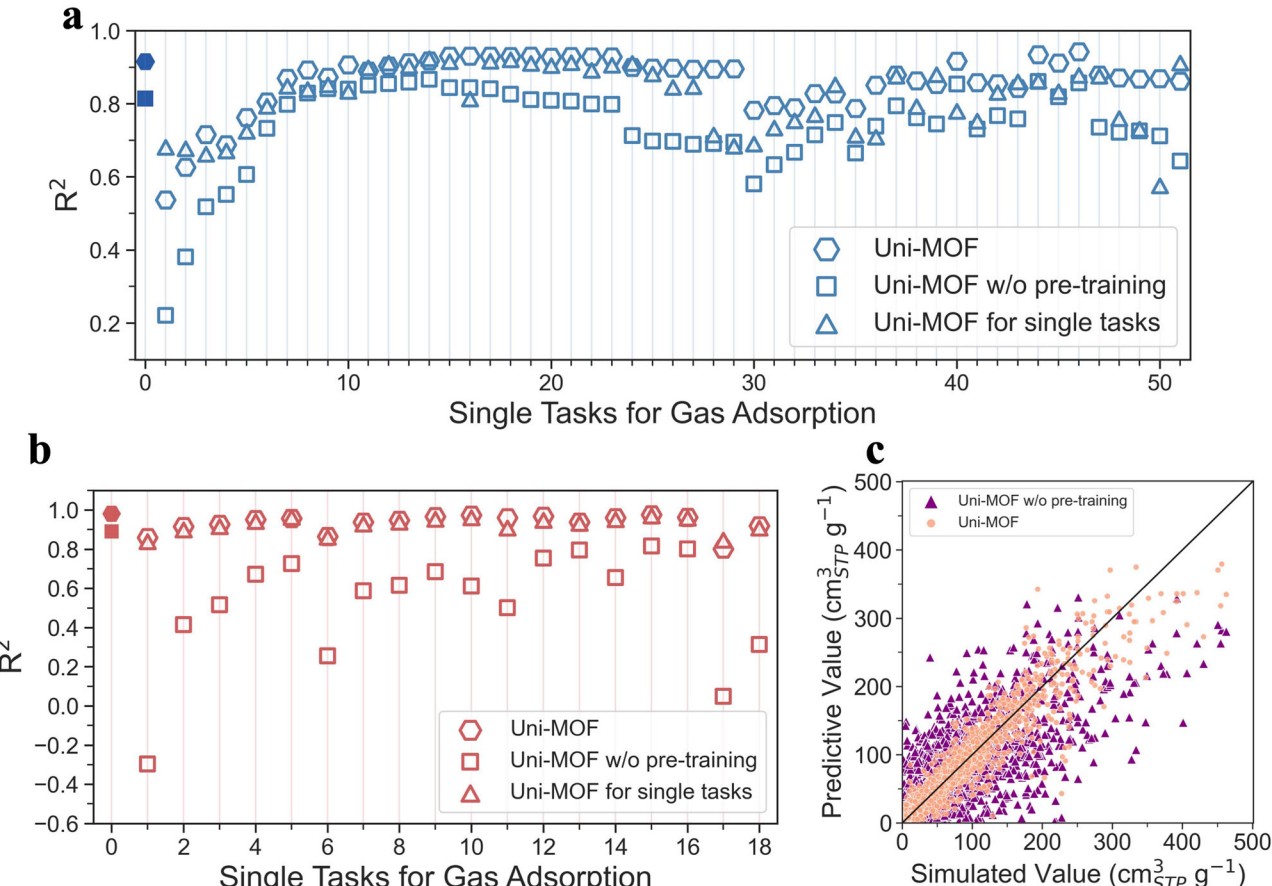

**Fig. 5 | Uni-MOF and Uni-MOF w/o pre-training comparison.** The performance comparison of Uni-MOF and Uni-MOF w/o pre-training in (**a**) CoRE_MOFX_DB and (**b**) hMOF_MOFX_DB. The filled marker represents the prediction performance of the entire database. The hollow marker represents the prediction performance of the sub-dataset (certain gas, temperature and pressure). Details can be found in Supplementary Table 15–17. **c** The comparison of correlation between predicted and simulated value of gas adsorption amount via Uni-MOF and Uni-MOF w/o pre-training for CoRE_MAP_DB database. $R^2$ means the coefficient of determination.

adsorption behavior for unknown gases is a formidable challenge. Variations in molecular size, surface adsorption energy, and intermolecular forces among different gas types have a significant impact on the adsorption mechanism and behavior. Despite this complexity, Uni-MOF exhibits exceptional generalizability, as evidenced by its ability to accurately predict the adsorption properties of unknown gases with only the support of adsorption data from other gases.

Furthermore, we investigated the capability of Uni-MOF to predict the gas adsorption behavior in COFs. Around 500 data points of $CH_4$ adsorption uptake in CoRE COFs at 300 K were simulated. Despite the limited size of the COF adsorption database, Uni-MOF maintains high predictive performance, achieving an $R^2$ of 0.76. Additionally, as shown in Fig. 4h, Uni-MOF exhibits high ranking capability for the adsorption of diverse materials under various pressures.

**Framework applying pre-training**

In order to verify that the self-supervised learning strategy of pre-training can effectively improve the robustness and downstream prediction performance of the Uni-MOF, we established and compared Uni-MOF w/o pre-training against Uni-MOF on three databases, namely CoRE_MOFX_DB, hMOF_MOFX_DB, and CoRE_MAP_DB. The CoRE_MOFX_DB and hMOF_MOFX_DB databases exhibit a high degree of data concentration, thereby enabling their partitioning into smaller databases containing adsorption data for various materials under identical working conditions (i.e., same gas, temperature, and pressure). We trained both CoRE_MOFX_DB and hMOF_MOFX_DB databases using Uni-MOF and Uni-MOF w/o pre-training, and subsequently

calculated the coefficients of determination for each small dataset. In addition, each small dataset is trained separately using the Uni-MOF framework to derive the corresponding coefficient of determination.

As shown in Fig. 5a, the filled markers represent the predictive performance of the whole CoRE_MOFX_DB database. It can be seen that the Uni-MOF performs better than Uni-MOF w/o pre-training. The self-supervised learning strategy of pre-training allows the model to learn the three-dimensional configuration of nanoporous materials in depth, thus improving the accuracy of model fine-tuning. In this part, the data sets are divided in a manner where each set comprises the relevant subsystem data. Therefore, all the hexagonal markers in Fig. 5a represent the overall predicted performance of Uni-MOF. Indeed, Uni-MOF demonstrates equally high performance on the complete data set as it does on an individual data set, even when the performance of a single data set relies on more extensive data than its specific prediction task. For fine-tuning of single-system tasks, that is, training individually for each small data set, the predicted performance hardly exceeded the performance of Uni-MOF. Thus, we know that far-ranging data sampling can further promote the prediction capacity of the learning model. The same conclusions can be summarized for hMOF_MOFX_DB from Fig. 5b.

However, due to the scattered sampling of the CoRE_MAP_DB database, limited small data sets cannot be divided following the same procedure as the previous two databases. Therefore, the correlations between predicted and simulated values from Uni-MOF and Uni-MOF w/o pre-training are compared, shown in Fig. 5c. Similarly, compared with Uni-MOF w/o pre-training, the performance of Uni-MOF was

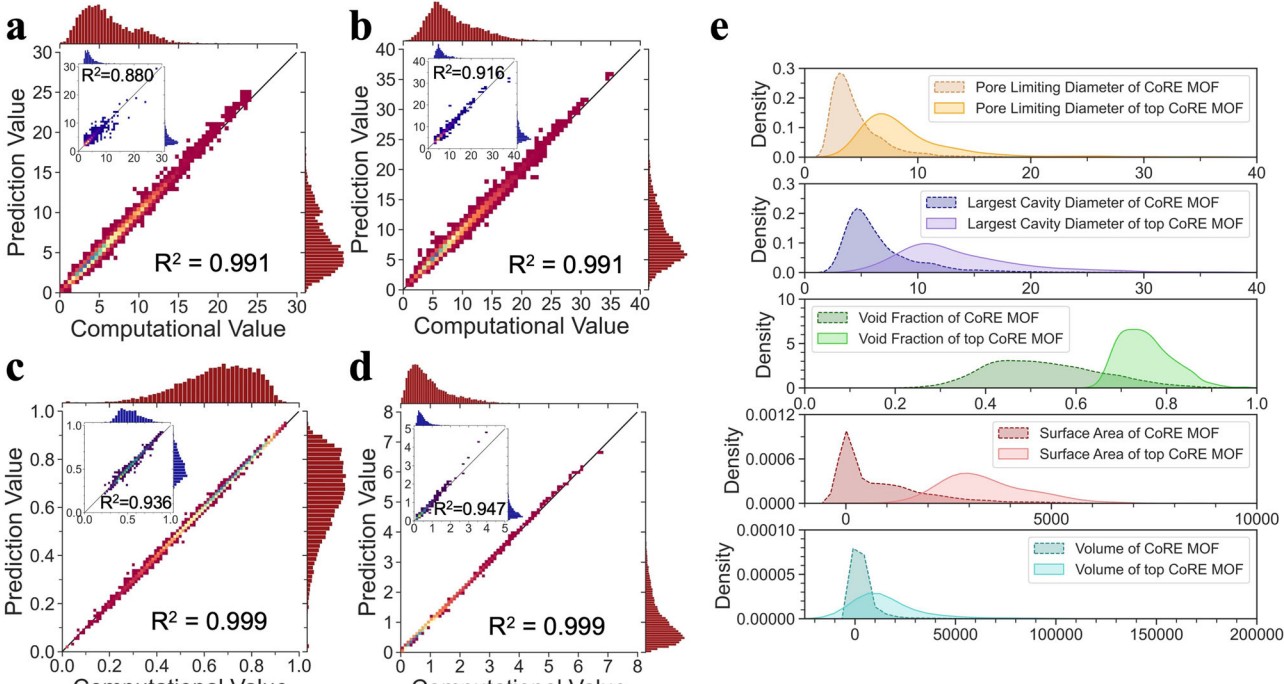

**Fig. 6 | Structural features prediction and analysis.** The correlation between predicted and computational value of (**a**) Pore Limiting Diameter (PLD) (Å), (**b**) Largest Cavity Diameter (LCD) (Å), (**c**) void fraction, and (**d**) pore volume (cm³ g⁻¹) of MOFs in hMOF (main figure) and CoRE_MOF (sub-figure) databases. **e** Comparison of the kernel density estimate (KDE) for different structural features [PLD (Å), LCD (Å), void fraction, surface area (m² g⁻¹), volume (Å³)] between the CoRE_MOF with all argon adsorption values and the CoRE_MOF with the top 10% performance of argon adsorption at 87 K and 1 bar. $R^2$ is the coefficient of determination, MOF means metal-organic framework. Source data are provided as a Source Data file.

---

improved from 0.70 to 0.83, which further proved the significance of the pre-training strategy for Uni-MOF.

**Structural features prediction and high-throughput screening**
While the Uni-MOF framework is adept at discerning the spatial arrangement of nanoporous materials, we aim to further explore its potential in forecasting structural attributes. The structural feature prediction results of Uni-MOF for the hMOF and CoRE_MOF materials libraries are presented in Fig. 6a–d.

It can be observed that the prediction performance of hMOF structural parameters is generally superior to that of CoRE_MOF. By employing a combination of structural building blocks (metal nodes, ligands, and functional groups), the design space of the hypothetical MOF (hMOFs) dataset was fully explored, and structural features are more evenly distributed compared to the CoRE_MOF dataset. Experimentally synthesized MOF materials (CoRE_MOF) tend to have structural preferences for metal nodes (e.g., with zinc, copper, and cadmium) and linkers. Previous research has demonstrated that the experimental MOF (CoRE-2019) is primarily concentrated in the small pore region, while the hMOF (Tobacco) is more evenly distributed with a slight shift of focus towards the large pore region[47]. The same observation can be found in Supplementary Fig. 6, where Supplementary Fig. 6a and Supplementary Fig. 6b show the differences between CoRE_MOF and hMOF structures using the t-distributed stochastic neighbor embedding (t-SNE) method. Through Supplementary Fig. 6c–f, it can be seen that the apertures of CoRE_MOF are obviously concentrated in the small aperture region, especially the PLD. hMOF, on the other hand, can still observe the aperture gradient due to its rich structural sampling and data points, even though the apertures are generally small. Thus, for different types of databases, the prediction of structural features performs differently, especially for PLD. Notably, in materials such as hMOFs, its predictive capability attains a coefficient of determination greater than 0.99, signifying a high level of

precision and dependability. One has come to recognize that Uni-MOF demonstrates a strong capability in predicting the structural features of materials, owing to the utilization of pre-training on a substantial number of three-dimensional structures. Thus, Uni-MOF not only accurately predicts the desired performance of MOF materials but also precisely predicts their structural features, which holds significant importance for material research and application.

Innumerable nanoporous materials can be created with varied secondary building units[48], leading to exceptionally diverse MOF structures, making MOF structure-property analysis a very strategic initiative. In this work, multi-gas adsorption uptakes under various operating conditions were collected and generated, where the argon uptakes at 87 K and 1 bar is representative and analyzable. Argon accounts for the vast majority of noble gases in the atmosphere (9340 ppm at ambient conditions), and has been widely used for insulation and illumination with a commercial value of 3.1 USD kg⁻¹[114]. Figure 6e shows a comparison of the kernel density estimate (KDE) for different structural features between the entire CoRE_MOF with argon adsorption values and MOFs in CoRE_MOF with the top 10% performance of argon adsorption. KDE visualizes the distribution of data using a continuous probability density curve in a less cluttered way.

As the distribution of some structural features is bounded, it may lead to distortions (such as the minus value of surface area and volume). However, it still presents interpretable and accurate trends in the structure of top MOF adsorbents. The typical parameters describing pore sizes are as follows. 1) pore limiting diameter (PLD) is the largest free sphere; 2) LCD is the largest included sphere along the free sphere path. Thus, LCD is intrinsically larger than PLD in the same material. Compared with the entire MOF database, the top 10% of MOFs have larger distribution values of 5–10 Å and 10–12 Å for PLD and LCD, respectively. The void fraction of the whole database is moderately distributed around 0.5, while that of the top 10% MOFs is much larger, mainly distributed around 0.75. Since most of the adsorption

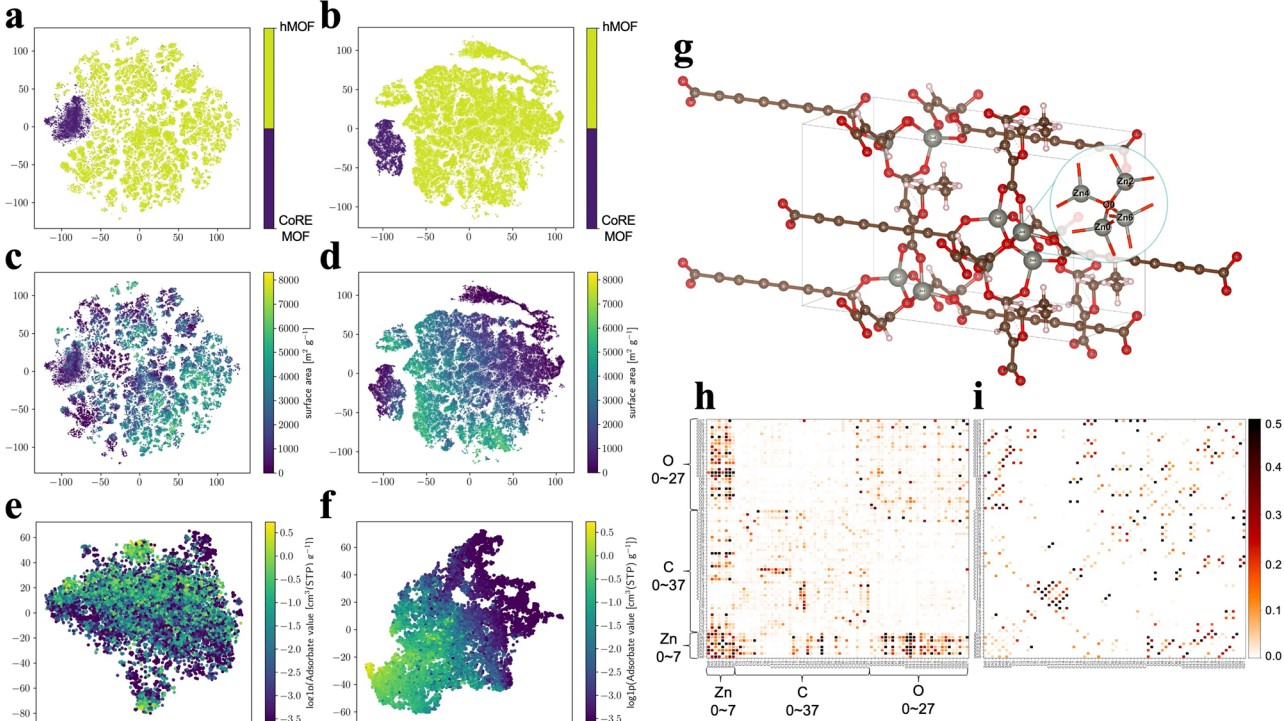

**Fig. 7 | Visualization of structural representations of MOFs in the hMOF and CoRE_MOF datasets, the low-dimensional embeddings are computed by t-SNE (t-distributed stochastic neighbor embedding) method.** The representations retrieved after (**a**), (**c**), and (**e**) pre-training and (**b**), (**d**), and (**f**) fine-tuning versus the other properties. **a, b** The low-dimensional embeddings versus dataset labels, where darker color represents CoRE_MOF and lighter color represents hMOF. **c, d** illustrate the representations versus surface area in m² g⁻¹ for hMOF and CoRE_MOF combined. **e, f** show the relationship of representations with respect to the adsorbate values of Ar at 87 K, 0.01 Pa for the CoRE_MOF dataset only. Explainable Artificial Intelligence (XAI). **g** Illustration of hMOF-5004238 structure and (**h**), (**i**) heat map of atomic interactions learning from multi-head attention algorithm in head 10 and head 18. MOF means metal-organic framework. Source data are provided as a Source Data file.

occurs on the surface of nanoporous materials, the surface area becomes one of the most critical determinants, and the results show that the top 10% of MOFs possess a very large surface area of about 3000 m² g⁻¹, which is consistent with common sense. The surface area determines the surface adsorption process, while the LCD determines the internal absorption process. Therefore, the top 10% MOFs were classified into three tiers according to their adsorption performance on argon, and the numerical distribution of these two key factors (surface area/LCD) was explored (shown in Supplementary Fig. 8). One can see that at 87 K and 1 bar, the most promising MOF adsorbents for argon mainly possess an LCD of about 15 Å and a surface area of about 4100 m² g⁻¹.

### Modeling of material structural representation
To validate that the MOF structures are well learned in both the pre-training and fine-tuning stages, we visualize the structural features, which are 512-dimensional vectors, using the t-SNE method[49]. The results are shown in Fig. 7. As illustrated in Fig. 7a, b, the learned features are capable of classifying MOFs either from CoRE_MOF or hMOF datasets. One may also notice that the boundary between CoRE_MOF and hMOF in the fine-tuned features is much more obvious, suggesting a significant improvement of these features after the fine-tuning. Remarkably, this is clearly demonstrated when we draw the embeddings versus the surface areas of MOFs, as shown in Fig. 7d, where the structures with small surface areas are located in the upper-right and lower-left corners, and the structures with large surface areas are in the center. In comparison to the pre-training features, Fig. 7c, the fine-tuning can significantly improve the representation quality.

On the other hand, we are more curious about whether the learned structural representations are closely correlated with the adsorption behaviors. To address this issue, we visualize the structural embeddings versus the adsorbate values of both Ar and N₂, and show the result of Ar here as an example, see Fig. 7e–f. As one can observe, the representations learned in the pre-training stage are not able to classify the structures with different adsorbate capacities while structures with various adsorbate values are grouped together, see Fig. 7e. However, after the fine-tuning, the structures with different adsorbate behaviors are well separated, demonstrating a good relationship between the learned representations and the target of adsorbate values, see Fig. 7f. This well explained the functions of the fine-tuning stage in further affecting the structural representations as well as other model parameters.

Furthermore, the multi-head attention mechanism of the Transformer can learn the interactions within the material structure. With the 64-Heads attention algorithm, the atomic interaction landscapes of hmof-5004238 in two different heads are represented in Fig. 7h, i. As shown in Fig. 7h, strong interactions can be observed between the metal sites (Zn), Zn and O atom, and also O atoms. Figure 7i also depicts the interaction between the linear carbon chains (C7-C15). In addition, there is a noticeable correlation between the O0 atom and the Zn atoms (Zn0, Zn2, Zn4, Zn6). The structural illustration in Fig. 7g further confirms this result, as the four atoms are chemically linked. Beyond this, the various chemical landscapes of hmof-5004238 in different heads are depicted in Supplementary Fig. 9. In this manner, Uni-MOF identifies tremendous materials and provides reliable predictions for diverse properties.

### Conclusion
In this study, we introduced Uni-MOF, a multi-purpose framework that can accurately predict gas adsorption in MOF materials. We also

generated, collected, and organized relevant databases of nanoporous materials and gas adsorption datasets. The self-supervised learning of a database containing over 631,000 MOFs and COFs was performed, resulting in a high prediction accuracy of 0.98. This indicates that the representation learning framework based on three-dimensional pre-training effectively learns the complex structural information of MOFs while avoiding over-fitting. We applied Uni-MOF to predict the gas adsorption performance of three major databases and achieved a high prediction accuracy of up to 0.98 in database with sufficient data. In the case of a sufficiently sampled dataset, Uni-MOF not only maintains a predictive accuracy above 0.83, but also accurately selects high-performance adsorbents under high pressure by solely predicting adsorption under low pressure, consistent with experimental screening results. Thus, Uni-MOF represents a significant breakthrough in the field of material science with regards to the application of machine learning techniques. Furthermore, our Uni-MOF framework shows superior performance on cross-system datasets compared to single-system tasks and can accurately predict the adsorption properties of unknown gases with a high prediction accuracy of up to 0.85, demonstrating its strong predictive ability and generality. Through extensive pre-training of three-dimensional structures, Uni-MOF effectively learns the structural features of MOFs, achieving a high coefficient of determination of 0.99 for hMOFs. Additionally, t-SNE analysis confirms that the fine-tuning stage can further learn structural features, and structures with different adsorbate behaviors are well identified, indicating a strong correlation between learned representations and gas adsorption targets.

Our database encompasses a pressure range of 0–10 bar and temperature ranges of 77–87 K and 150–300 K, making it suitable for the majority of gas adsorption issues. Uni-MOF has demonstrated precise prediction outcomes for a fairly comprehensive collection of MOF structures and distinct COF materials. Even under some extreme conditions, the predicted trend remains highly dependable. With Uni-MOF, the continuous updating of databases and even usage scenarios are supportable, which further emphasizes its universality. In summary, Uni-MOF framework serves as a versatile predictive platform for MOF materials, functioning as a gas adsorption estimator for MOFs, as it exhibits high precision in predicting gas adsorption under diverse operating conditions and has broad applications in the field of material science.

## Methods

### Materials and data collection and generation

The MOF/COF structures used for pre-training are either collected from the currently available database or generated using the corresponding program. There is a wealth of existing MOF/COF databases, including computer-synthesized databases of hMOFs[50], ToBaCCo (Topologically Based Crystal Constructor) MOFs, and experimental-level databases of CoRE (Computation-Ready Experimental) MOFs[51], CoRE COFs[52] and CCDC (The Cambridge Crystallographic Data Centre), etc. One integrated database online is MOFXDB, where more than 168,000 MOF/COF structures are available. Apart from exploring nanoporous materials in the materials library, we employed the ToBaCCo.3.0 program to generate over 306,773 MOF structures. The ToBaCCo program takes as input a topological blueprint, searches compatible node building blocks from a defined set, and constructs all possible structures using the topology in combination with different node building blocks and edge building blocks. In particular, the program generates structures with three folders as input variables, i.e., "templates", "nodes", "edges". To generate as many MOF structures as possible, we use all edges as provided in "edges_database", all nodes as provided in "nodes_database" and all templates as provided in "template_database". For the downstream task, i.e., gas adsorption uptake by MOFs, we collected data from online sources such as MOFXDB, composing datasets of more than 2,400,000 sorptions of hMOFs on

five gases ($CO_2$, $N_2$, $CH_4$, Kr, Xe) at 273/298 K and 0.01–10 Pa and over 460,000 sorptions of CoRE MOFs on two gases (Ar, $N_2$) at 77/87 K and 1–$10^5$ Pa. In this work, only single component gas adsorption data are considered. In addition, we conducted Grand Canonical Monte Carlo (GCMC)[53] simulations using the RASPA[54] software to produce another 99,000+ gas adsorption uptake dataset, with 50,000 initialization cycles and an additional 50,000 cycles employed for adsorption capacity samples. The collected sorptions were obtained within 150–300 K and 1 Pa–3 bar, considering seven types of gas molecules ($CH_4$, $CO_2$, Ar, Kr, Xe, $O_2$, He). Interactions between gas molecules and atoms in adsorbent materials were described in terms of Lennard-Jones (12–6) potential, and the cutoff radius is set to 12.9 Å with the tail-correction. The force field parameters of framework atoms were described by UFF (Universal Force Field)[55]. The force field parameters of noble gas molecules (i.e., argon, krypton, and xenon) were estimated from the principle of corresponding states for the second virial coefficients[56] (listed in Supplementary Table 19). The molecular model Transferable Potentials for Phase Equilibria[57] was also used. In our work, the partial charge of the framework atoms is not considered because of computational cost and significant deviations observed in adsorption results by using different partial charges assignment methods[58].

### Material analysis

The key to high-throughput computational materials science is robust software tools. Here, we use the Python Materials Genomics[59] (pymatgen), a robust and open-source Python library, to derive useful material properties from raw crystallographic structural data and conduct comprehensive materials analysis. Materials properties, including lattice vectors, lattice angles, unit cell volume, atoms, and coordinates, are extracted using the pymatgen. The atom types and coordinates will be used in the pre-training stage for the self-supervised learning strategy. The Python package - OpenMetalDetector[51], was used to analyze collections of Metal Organic Frameworks for open metal sites.

### Uni-MOF framework

**Pre-training.** We employed Uni-Mol as the pre-training framework, which is a dedicated pure three-dimensional pre-training framework designed for molecules. Uni-Mol has shown high performance in various downstream tasks in the field of drug discovery. However, due to the completely different structure and three-dimensional spatial distribution of MOF materials compared to small molecule drugs, as well as the periodic boundary conditions (PBC) of porous structures and the much larger number of heavy atoms in crystals, we made necessary modifications to Uni-Mol based on these facts.

1) We leverage an extra head of the lattice matrix to preserve cell geometric information. The presence of PBC is considered in MOF representation learning as PBC is natural for MOF materials. Besides masked atoms prediction and coordinates recovery in Uni-Mol pre-training, a regression head to prediction lattice 3 × 3 matrix is used to learn the PBC information.

$$\mathcal{L}_{lattice} = MSE(A, \hat{A}) = \frac{1}{n}\sum_{i=1}^{n}|A_i - FFN(CLS^i_{repr})|^2 \qquad (1)$$

$$\mathcal{L} = \mathcal{L}_{lattice} + \mathcal{L}_{Uni-Mol} \qquad (2)$$

where $CLS_{repr}$ indices representation of Uni-Mol for classification token, classification token ($CLS$) refers to the special token in the input sequence of atoms, which is used to represent the entire molecule in the output of Uni-Mol. $A$ is the lattice matrix, we use Feedforward Neural Network ($FFN$) of $CLS_{repr}$ to predict lattice matrix with optimizing MSE loss directly. $\mathcal{L}$ in Uni-MOF pre-training is a

summation of $\mathcal{L}_{lattice}$ and original $\mathcal{L}_{Uni-Mol}$. In $\mathcal{L}_{Uni-Mol}$, the atom-masked prediction and coordinates recovery are two major components of loss items. For more details, the readers are welcome to refer to original paper of Uni-Mol.

2) We propose an edge gated kernel for geometric spatial positional encoding. MOFs share totally different arrangement of atoms compared to drug molecules in three-dimensional space, with porous structures and an average above 1000 heavy atoms in a single cell. Uni-Mol uses a Gaussian kernel to encode spatial positional information, while in MOF pre-training suffers from training instability with a Gaussian kernel of much larger pair distance and atom counts. To address this, we propose an edge gated distance kernel.

$$\mathcal{A}(d, r; \boldsymbol{a}, \boldsymbol{b}) = a_r d + b_r \qquad (3)$$

$$\boldsymbol{p}_{ij} = \boldsymbol{p}_{ij}^{proj} + \boldsymbol{p}_{ij}^{emb} = LN(FFN(d_{ij})) + \sigma(\mathcal{A}(d_{ij}, t_{ij}; \boldsymbol{a}, \boldsymbol{b})) \cdot Embedding(t_{ij}) \qquad (4)$$

where $d_{ij}$ is the Euclidean distance of atom pair $ij$, and $t_{ij}$ is the edge type of atom pair $ij$. Please note the edge here is not the chemical bond, and edge type is determined by the atom types of pair $ij$. $\mathcal{A}(\cdot, \cdot; \boldsymbol{a}, \boldsymbol{b})$ is the affine transformation with parameters $\boldsymbol{a}$ and $\boldsymbol{b}$, it affines $d_{ij}$ corresponding to its edge type. $p_{ij}$ indices the edge gated distance kernel, which is a summation of distance projection and edge gated embedding. FFN is the Feedforward Neural Network about the non-linear transformation of distance $d_{ij}$, and LN is the LayerNorm operation. A sigmoid gated is used as the affine transformation of $\mathcal{A}$ to weight edge pair type embedding.

**Fine-tuning.** Prediction of multi-gas adsorption under different operating conditions, the fine-tuning model should be fed with not only the three-dimensional spatial structure but also the gas and operating conditions (i.e., temperature and pressure). Therefore, gas block and temperature/pressure blocks are proposed in Uni-MOF to form a cross-system performance prediction module.

*Gas block* The gas representation is a combination of gas id and gas intrinsic property-related descriptors.

$$\hat{x}_g = concat(Eembedding(g_i); FFN(g_x)) \qquad (5)$$

where $g_i$ indices the gas id of gas $g$, $g_x$ represents the gas $g$ descriptors (listed in Supplementary Table 2). The gas representation $\hat{x}_g$ is a concatenation of gas id embedding and FFN layer mapping of gas descriptors.

*Num block* We use Equal Distance Discretization (EDD) and Logarithm Discretization (LD) for temperature and pressure encoding, respectively. EDD first maps the numerical value into the corresponding bucket with equal width, then applies embedding mapping. LD utilizes the logarithm transform with EDD to accommodate logarithmic likely features.

$$EDD(x_j) = Eembedding(\lfloor((x_j - x_j^{min})/w)\rfloor) \qquad (6)$$

$$LD(x_j) = EDD(\log_{10}(x_j)) \qquad (7)$$

$$\boldsymbol{x}_{num} = concat(\boldsymbol{EDD}(x_{temp}); \boldsymbol{FFN}(x_{temp}); \boldsymbol{LD}(x_{pressure}); \boldsymbol{FFN}(x_{pressure})) \qquad (8)$$

where the interval width is noted as $w = (x_j^{max} - x_i^{min})/N_j$, $x$ denotes numerical features, $x_{temp}$ and $x_{pressure}$ are the original temperature and pressure values of the corresponding environment. For the temperature feature, we use EDD and FFN embeddings, and for the pressure feature, we choose to use LD and FNN with consideration of logarithmic likely transformation in pressure.

## High-throughput analysis of large MOF database
To further investigate the effect of material structure on gas adsorption, Zeo++[60], a software package for crystalline porous materials analysis, was used to perform an analysis of the structure and topology of the material geometry. Structural features include the LCD, PLD, void fraction, void volume, and specific surface area. The internal void volume can largely influence the performance of nanoporous materials. Precise definitions exist for volume from different methods, especially the accessible and nonaccessible probe center pore volume (Ac-PC, NAc-PC), accessible and nonaccessible probe-occupiable pore volume (Ac-PO, NAc-PO)[61]. The concept of accessibility relies on the probe size, and a probe of the radius of 1.8 Å is used in our work in order to directly relate the pore volumes to experimentally measured Nitrogen (kinetic diameter of 3.64 Å) isotherms. Accessible volume calculated here is also defined as the volume available to the center of the spherical probe, corresponding to the accessible probe center pore volume.

## Statistical data visualization
In this work, Python data visualization libraries such as seaborn[62] and matplotlib[63] were used for informative statistical graphics. Three-dimensional structures are drawn using the web service Bohrium™ at https://bohrium.dp.tech.

## Computing environment
Uni-MOF pre-training and fine-tuning are performed on V100/A100 GPUs, and Monte Carlo simulations are performed on CPU cluster of Bohrium™.

## Reporting summary
Further information on research design is available in the Nature Portfolio Reporting Summary linked to this article.

# Data availability
The data presented in this study are available in the manuscript file, the Supplementary Information file and Source Data files. Source Data in this study have been deposited in the figshare database[64] under accession code of https://doi.org/10.6084/m9.figshare.24996317.

# Code availability
Code to run the Uni-MOF model is available in GitHub (https://github.com/dptech-corp/Uni-MOF)[65].

The notebook demo can be found at https://bohrium.dp.tech/notebook/cca98b584a624753981dfd5f8bb79674.

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

## Acknowledgements

D.L. was supported by National Natural Science foundation of China, No. U1862204 and 21878175. This research was supported in part by the Tsinghua University and the DP Technology.

## Author contributions

J.W.(Wang) and J.L. contributed equally to this work. D.L., Z.G. and J.W.(Wu) designed and guided the project. J.W.(Wang), J.L. and Z.G. performed the data collection, developed the breakthrough model and drafted the paper. H.W., M.Z., G.K. and L.Z. provided important advice and revised the paper. All authors contributed to the discussion of results and commented on the paper.

## Competing interests

The authors declare no competing interests.
