## [Peer Review File · Nature Communications]

A Comprehensive Transformer-based Approach for High-Accuracy Gas Adsorption Predictions in Metal-Organic FrameworksREVIEWER COMMENTS

Reviewer #1 (Remarks to the Author):

The manuscript basically describes the development of a machine-learning (ML) model called Uni-MOF that can predict structural features of metal-organic frameworks (MOFs) and various gas uptakes in them at different conditions with varying accuracies. The Uni-MOF model can predict structural features of MOFs highly accurately. The gas uptake prediction accuracies of the Uni-MOF with pre-training are, in general, satisfactory for a wide variety of gases at different temperature and pressure conditions. The use of the developed model could result in significant savings in computational time as it could determine potentially high-performing MOFs in a very short period as opposed to long simulations. Therefore, the presented model holds much promise to accelerate the discovery of high-performing MOFs for gas storage. As the community of gas capture/storage in MOFs expands rapidly, the manuscript could be of interest to many researchers from different disciplines working on the storage of different types of gases using experimental and/or computational techniques. The references cited are recent and appropriate. In general, the manuscript conveys its message clearly. Yet, there are some main aspects, including those about validity, originality, and significance of the work indicated below, that should be addressed before the manuscript may be published.

- 1) On page 2, it is mentioned that Uni-MOF is the first of its kind. Please clearly mention the unique and pioneering features of Uni-MOF and describe the superior aspects of the developed model over previously developed Transformer-based models, MOFormer (<https://doi.org/10.1021/jacs.2c11420>) and MOFTransformer (<https://doi.org/10.1038/s42256-023-00628-2>), if there is any?
- 2) It is mentioned that the model development involves not only the use of MOF structures but also COFs. What are the advantages of including COF structures in the model training over training the model with only MOF structures? Did the authors check whether involving COFs in the model training could worsen the predictions of structural features and/or gas uptakes?
- 3) Do the authors think that the developed model is transferable and can be used for gas uptake predictions for COFs as well? If not, what extra information would be needed to incorporate into the model?
- 4) In Fig. 2a, overall, while the model predictions represent simulated values well, there are also many points that appear to be systematically deviating from parity line especially at high uptake values. This gives the impression that the developed model could provide good predictions overall, but it may not serve as a good predictor model for individual MOF structures. Could the authors please discuss to what extent the material rankings could differ based on simulations and model predictions for different gas uptakes and databases? Also, what pathways could be followed to avoid this systematic error while developing the model?
- 5) In Fig. 2a, it is observed that while the simulated values keep increasing beyond 9-10 mol/kg, the predicted maximum uptake value remains around 9-10 mol/kg. This suggests that if the developed model was used to predict the gas uptake values in high performing MOFs (with simulated uptake values beyond 14 mol/kg), it would not be able to provide realistic uptake values making it hard to identify the best performing MOFs. This implies that the use of the developed model for a large MOF database may help identify large groups of high and low-performing MOFs but not lead to quantitative differences in terms of gas storage performance for high-performing MOFs as the predicted values would be very similar. Given the excessive number of MOFs generated so far (in the

order of hundreds of trillions, <https://doi.org/10.1021/acsami.1c02471>), the use of the developed model for large MOF databases could result in having excessively long lists of structures rather than “shortlists” for which performing computational simulations could be practical. Considering such a case, what strategy could be implemented to differentiate the performance disparities across many high-performing MOFs that could be identified by the developed model?

6) On page 4, the authors indicate that “The model with the highest validation set R2 (coefficient of determination) was saved as the optimal model, and the final R2 was reported using the weights of this model to reasonably avoid over-fitting.” Considering this sentence, it is not quite clear how the overfitting was avoided. Could the authors please provide more details on how they ensured the robustness of model and reduced the risk of overfitting?

7) In Fig. 3, many of the predicted values at low pressure are considerably away from the experimental values especially for Mg-dobdc and MOF-5. It is known that simulations may not reproduce experimental values especially when the MOF involves open-metal sites as in Mg-dobdc. However, MOF-5 has closed metal sites and it was shown earlier that simulations could describe the experimental gas uptake values well (<https://doi.org/10.1021/am506793b>). Given the fact that the developed model relies on simulations which can reproduce experimental values for structures like MOF-5, what could be the reason(s) for having highly inaccurate predictions at low pressure for CO₂ and CH₄ uptake in MOF-5?

8) On page 6, it is mentioned that experimental values were used to correct the fitted adsorption isotherms. However, no detail is provided about this correction. Could the authors please elaborate on the details of adsorption isotherm correction?

9) In Fig. 4g, it is shown that predictions of Kr uptake are more accurate than those of other gas uptakes. It would be great if the authors can provide reasoning for the entire observed trend. For instance, it is interesting that the gas uptake predictions for Xe, which has similar characteristics with Kr, are much less accurate while the prediction accuracy for Ar (another similar inert gas) uptake is somewhere between those of Kr and Xe uptakes. Could the authors please suggest any physical/chemical reason for that?

10) In Fig.5, considering the cases where the ML models were trained individually for small data sets, the prediction accuracies of some of those cases are higher than those of Uni-MOF which was trained with the entire data set. Could the authors please elaborate on the underlying reasons for this observation?

11) In Fig. 6a, it is shown that the predicted PLD values for CoRE MOFs (especially with large PLDs) could be considerably away from the parity line while those for the hMOFs are very close to the parity line. Could the authors explain why the PLD predictions for synthesized MOFs could be less accurate than those for the hypothetical MOFs?

12) What SBUs were selected for the generation of more than 300,000 MOFs and what were the reasons to select those SBUs?

13) Could the authors please discuss the limitations of the developed model? For instance, how safe would it be to get gas uptake predictions with the developed model at the unexplored conditions (physically/chemically different MOFs, temperature/pressure out of the bounds investigated)? In short, do the authors think that the developed model is fully universal for the adsorption of gases investigated in MOFs?

There are also a number of minor issues mentioned below that needs to be handled.

1) On page 4, it is mentioned that the data set was divided into 8:1:1 which very likely refers to ratios of training, validation, testing sets, respectively. However, this is not explicitly mentioned. Could the authors please mention the sizes of those sets in the text to be clearer? Also, in the same sentence the authors mention that the data set was not randomly split. Could the authors elaborate

on it more to help readers better understand how exactly the data splitting was made (e.g., stratified sampling)?

2) In Fig. 3, it seems lines represent Langmuir fits. Please indicate it in the figure caption for clarity if that is the case.

3) In the caption of Fig. 4, it is mentioned that “The adsorption isotherms are obtained only by simulated value.” however the caption also states that “The lighter color dot represents the predicted adsorption uptake of Uni-MOF” which makes the first statement confusing as it is implied that adsorption isotherms were obtained using ML model and simulations. Did the authors mean the adsorption isotherms were simulation-based and not experiment-based? If that is the case, the first statement seems redundant. Could the authors please clarify it?

4) In Fig. 4a-f, do different markers denote different temperatures? If yes, it would be better to add legend(s) mapping marker shapes to temperatures. Also, it is not clear in those subplots how gray lines were plotted? Looking at Fig. 4e, the lines do not go through all markers which suggests that those lines were not drawn to guide the eye.

5) On page 12, it is mentioned that void fraction, void volume, and surface area were obtained using Zeo++. Please provide probe sizes used for those computations and specify whether these computations rely on the concept of probe-occupiable volume or probe center pore volume as defined in <https://doi.org/10.1021/acs.langmuir.7b01682>

6) Not much simulation details were provided for GCMC simulations such as atomic partial charges, forcefields, molecular models etc. To enable reproducibility of the work, please provide computational details as necessary. Also, on page 11, it is stated that 50,000 “steps” were used as initialization “cycles” and additional 50,000 steps were used for production. Since steps and cycles mean two different things in RASPA, could the authors please clarify whether they used 50,000 steps or cycles for initialization and production?

7) On page 8, it is mentioned that Argon accounts for the vast majority of the atmosphere while its concentration in the atmosphere is actually low, which is also stated in the same sentence. Did the authors mean Argon accounts for the majority of the inert gas content in the atmosphere?

8) In Fig. 6d and Table S10, it would be better to state the type of “volume” (e.g., pore volume, unit cell volume) to avoid any confusion.

9) On page 11, it is mentioned that pymatgen was used to extract material properties. Could the authors please be more specific on the material properties calculated using pymatgen? If that is a long list of properties, it could be provided in the Supporting Information.

10) In the Author Contributions section, it is mentioned that three authors conducted breakthrough experiments which is confusing as no breakthrough data were presented.

11) In the captions of Table S7 and S8, Uni-MOFsingle and Uni-MOFmixed notations were explained however they do not appear in the corresponding table. Instead, Uni-MOFI&SP and Uni-MOFMAP exist in the table which were not defined. Please clarify it.

12) MOFX-DB which is used as one of the data sources mainly involves single-component gas adsorption results as well as some multi-component gas adsorption results. It would be better if the authors could be more specific on whether the gas adsorption data used to train the models were results of single-component, multi-component gas adsorption simulations (or a combination of both).

13) What are the error margins in percentage for the simulated gas uptake values used for model training?

Reviewer #2 (Remarks to the Author):

In this work Wang and coworkers present a machine learning (ML) workflow and result for prediction of gas adsorption in MOFs. The most impactful part of the work is the Uni-MOF module which is capable of describing MOFs in a way that is amenable for use in other ML workflows and they demonstrate that it learns textural property information for the MOFs. However, there are claims made in the manuscript that I don't believe are substantiated with the presented data and thus I cannot recommend publication. Namely:

1. Claims as far as universality of adsorption predictions go are not really tested or demonstrated. Authors make claims of being able to make adsorption predictions at multiple conditions. However, the only data that is presented is some isotherms in some MOFs with at most 100 K difference in temperature of the isotherms. That is not enough to make a claim of universality of adsorption predictions.
2. Along the lines of universality and using data from one adsorbate to make predictions about others, authors should compare their work against the recommender system from Cory Simon and the alchemical isotherm ML predictions from Gomez-Gualdrón. Authors should comment on this and clearly express what are the advantages of the presented approach.
3. Another problem with their claim of universality is that they need to train a model for each database separately. If each database of structures requires separate training, then the claims of universality are not very strong.
4. Further issues with the claim of universality is the fact that they report a prediction accuracy of "above 0.35" for all unknown gases. That is not a good result, especially when compared to others from the literature that I've already highlighted.
5. Claims are made about predicting high pressure loading from low pressure loading. While it may be true, it is hardly an impressive result. Simple estimations of pore volume and gas density can provide accurate estimates of saturation loading in porous materials, which combined with Henry's constant or low pressure data can be used to fit a Langmuir isotherm and obtain similar results.
6. Authors mention that this is a "gas adsorption detector" for MOF materials. I am not sure I understand exactly what they mean by gas detector in this context. This should be clarified.
7. Authors report R2 values for the various models. Figure 2, panel a seems to have some systematic errors at the highest part of the graph where there seems to be a horizontal line. Authors should expand on this and explain that in detail. Further, this brings up the point that perhaps other error metrics like absolute error, mean absolute error, mean relative error, etc. should also be discussed.

Reviewer #3 (Remarks to the Author):

Recommendation: Publish after minor revisions

Wang and et. al. report on the development of the Uni-MOF. This graph neural network can accurately predict the gas adsorption of MOFs and COFs with various adsorbates at various experimental conditions, while only relying on the crystal structure for featurization. The authors demonstrate the utility of their models for finding high-performance adsorbents at high pressure, while only using low-pressure data in their training set. Because the model is trained for predictions with a wide variety of adsorbates, I believe that this work could have a significant impact on the gas

separation field and can be employed in high-throughput virtual screening workflows. The authors were thorough in their analysis, not only demonstrating that pre-training their models improves model accuracy but also highlighting that pre-trained features also uncover structural features of these materials, such as surface area. Furthermore, the authors demonstrate that the model doesn't simply rely on learning from adsorbate and experimental conditions, showing that the model accuracy doesn't degrade when focusing on a single adsorbate and a constant experimental condition, highlighting that MOF/COF learned features play an integral part in model's predictions. With that all being said, I have two comments I would like the authors to address.

In the section on cross-system forecasting, I believe that the authors are overstating the performance of their models when employing group splits using adsorbate identity. While performance with Kr is impressive, in other cases performance is mediocre and I'm not sure if any of the adsorbate features are playing any role in the predictions here, or if the model is instead predicting based on the general capabilities of different adsorbents and experimental conditions.

Finally, I think the quality of the work would be significantly enhanced if the authors could employ some feature importance analysis to get more chemical insights into factors influencing gas adsorption capacities, or differing adsorption capacities between adsorbates. The biggest drawback of GNNs is their black-box nature, and significant progress has been made in recent years toward more explainable AI, and some of these approaches could be implemented in this work to make this work more interesting to experimentalists.

Reply to Comments

Reviewer: 1

Recommendation: Major revisions needed as noted.

Comments: The manuscript basically describes the development of a machine-learning (ML) model called Uni-MOF that can predict structural features of metal-organic frameworks (MOFs) and various gas uptakes in them at different conditions with varying accuracies. The Uni-MOF model can predict structural features of MOFs highly accurately. The gas uptake prediction accuracies of the Uni-MOF with pre-training are, in general, satisfactory for a wide variety of gases at different temperature and pressure conditions. The use of the developed model could result in significant savings in computational time as it could determine potentially high-performing MOFs in a very short period as opposed to long simulations. Therefore, the presented model holds much promise to accelerate the discovery of high-performing MOFs for gas storage. As the community of gas capture/storage in MOFs expands rapidly, the manuscript could be of interest to many researchers from different disciplines working on the storage of different types of gases using experimental and/or computational techniques. The references cited are recent and appropriate. In general, the manuscript conveys its message clearly. Yet, there are some main aspects, including those about validity, originality, and significance of the work indicated below, that should be addressed before the manuscript may be published.

[Authors' Reply] We thank the referee for carefully reviewing this manuscript with constructive suggestions and builds. We added new results and analysis to further improve the quality of our work.

(1) On page 2, it is mentioned that Uni-MOF is the first of its kind. Please clearly mention the unique and pioneering features of Uni-MOF and describe the superior aspects of the developed model over previously developed Transformer-based models, MOFormer(<https://doi.org/10.1021/jacs.2c11420>) and MOFTransformer (<https://doi.org/10.1038/s42256-023-00628-2>), if there is any?

[Authors' Reply] We thank the referee for reminding us of discussing the uniqueness of Uni-MOF comparing with other Transformer-based models. UniMOF, as a transformer-based pre-training framework, not only directly recognizes the 3D spatial structure of the material and make model robust. It can also further consider the operating conditions (e.g., temperature, pressure, and gas type) to build more engineered machine learning models, which is a big step from scientific research to real-world applications, making Uni-MOF a very unique framework. Additionally, with Uni-MOF, the continuous updating of databases and even usage scenarios are supportable, making it a completely universal framework. We have therefore made the following revision.

(Page 2, paragraph 4) “Compared with other Transformer-based models such as MOFormer and MOFTransformer, Uni-MOF, as a Transformer-based framework, not only can the pre-training recognize and recover the three-dimensional structure of nanoporous materials and thus greatly improve the robustness of the model, but the fine-tuning task also further takes into account the operating conditions such as temperature, pressure, and different gas molecules, which makes Uni-MOF suitable for both scientific research and practical applications.”

(Page 12, paragraph 3) “Our database encompasses a pressure range of 0-10 bar and temperature ranges of 77-87 K and 150-300 K, making it suitable for the majority of gas adsorption issues. Uni-MOF has demonstrated precise prediction outcomes for a fairly comprehensive collection of MOF structures and distinct COF materials. Even under some extreme conditions, the predicted trend remains highly dependable. With Uni-MOF, the continuous updating of databases and even usage scenarios are supportable, which further emphasizes its universality.”

(2) It is mentioned that the model development involves not only the use of MOF structures but also COFs. What are the advantages of including COF structures in the model training over training the model with only MOF structures? Did the authors check whether involving COFs in the model training could worsen the predictions of structural features and/or gas uptakes?

[Authors' Reply] We thank the referee for pointing this out and helping us better clarify the model performance. Pre-training involving COF structures can help computers learn the structural features of different nanoporous materials, which not only further improves the prediction performance of MOF material related properties, but also expands the application scenarios of the model, such as downstream task prediction for COF materials. Here, we compared the prediction performance of two models (*i.e.*, pre-training stage using both MOF and COF or MOF alone). In the pre-training stage, the prediction accuracy of both cases reaches 0.98, which indicates that the model can learn the three-dimensional spatial structure of MOF and COF well. In the downstream task part, the prediction performance (whether R^2 , RMSE, or MAE) of Uni-MOF for all three databases exceeded that of Uni-MOF with only MOF pre-training. This means that the addition of COF not only does not reduce performance, but also makes the model more robust, which also shows the superiority of the Uni-MOF framework. We added some sentences to clarify above discussion.

(Supporting Information, Page 2) Table S3

(Supporting Information, Page 2) Table S4

(Supporting Information, Page 3) Figure S1

(Page 4, paragraph 1) “High-throughput construction of COFs based on materials genomics strategy with quasi-reactive assembly algorithms (QReaxAA) is feasible, leading to a comprehensive library of COFs. Through the spatial configuration of materials, Uni-MOF is capable of learning the material structural properties, most importantly the chemical bonding information, very well. In order to enable Uni-MOF to learn more diverse materials and thus improve the generalization ability to new materials, we introduced MOFs and COFs both virtually and experimentally during the pre-training process.”

(Page 5, paragraph 2) “Additionally, we discover that the prediction accuracy in the pre-training stage could reach 0.98 before and after incorporating COF materials. This indicates that Uni-MOF is capable of effectively learning the three-position spatial structure of multiple nanoporous materials. As for the downstream tasks, the predictive performance (R^2 , RMSE, and MAE) of Uni-MOF for all three databases surpassed that of Uni-MOF with only MOF pre-training, as illustrated in Figure S1. This indicates that incorporating COF not only maintains but also enhances the model robustness, further demonstrating the superiority of our Uni-MOF framework.”

(3) Do the authors think that the developed model is transferable and can be used for gas uptake predictions for COFs as well? If not, what extra information would be needed to incorporate into the model?

[Authors’ Reply] We thank the referee for considering the model applicability. In fact, Uni-MOF can predict adsorption properties in COF materials by simply providing CIF file like MOF materials. This is also our intention to include COF materials in the pre-training phase to make Uni-MOF a fully universal framework. We simulated the methane adsorption by the CoRE COF at 300K and gathered a dataset comprising approximately 500 data points. The findings indicate that the prediction performance of Uni-MOF achieves not only the R^2 of 0.76 but also exhibits a high degree of accuracy in ranking the adsorption isotherms. In the near future, the continuous updating of databases involving more COFs and even other nanomaterials, can make this model self-updating and more universal. We added the content to elucidate this.

(Page 8) Figure 4h

(Page 8, Figure 4 Caption) **h** Predicted and simulated CH₄ adsorption isotherms at 300 K in COF materials, different marker types represent different COFs.

(Supporting Information, Page 11) Table S12

(Page 7, paragraph 6) “Furthermore, we investigated the capability of Uni-MOF to predict the gas adsorption behavior in COFs. Around 500 data points of CH₄ adsorption uptake in CoRE COFs at 300K were simulated. Despite the limited size of COF adsorption database, Uni-MOF maintains outstanding predictive performance, achieving an R^2 of 0.76. Additionally, as shown in Figure 4h, Uni-MOF exhibits outstanding ranking capability for the adsorption of diverse materials under various pressures.”

(4) In Fig. 2a, overall, while the model predictions represent simulated values well, there are also many points that appear to be systematically deviating from parity line especially at high uptake values. This gives the impression that the developed model could provide good predictions overall, but it may not serve as a good predictor model for individual MOF structures. Could the authors please discuss to what extent the material rankings could differ based on simulations and model predictions for different gas uptakes and databases? Also, what pathways could be followed to avoid this systematic error while developing the model?

[Authors’ Reply] We thank the referee for the constructive suggestion. Here two other error metrics, mean absolute error (MAE) and root mean square error (RMSE), were added to the discussion of results. The hmof dataset demonstrates superior performance with the highest R^2 value, as well as remarkably low MAE and RMSE. Despite the CoRE_MAP dataset exhibiting a lower R^2 , as a result of extensive sampling and limited data volume, it also presents low MAE and RMSE values. This indicates that the error of outliers is acceptable in the hmof and CoRE_MAP datasets. However, for the CoRE_MOFX dataset, we can see some points deviating from parity line. CoRE_MOFX contains the adsorption amounts of Ar and N₂ at 77K and 87K, which are much lower temperatures than the other two datasets, making the adsorption values in CoRE_MOFX generally large. Here we describe the adsorption isotherms with the greatest errors in Figure S2. Interestingly, even with the largest errors, the predicted adsorption isotherm trends still match the original data points. At the same time, MOF materials with the greatest error generally have Cu (with the lowest energy parameter) as the metal site. This indicates that MOF materials with low energy parameter metal sites or very large pore size are the bottleneck for prediction at low temperature data. But even so, Uni-MOF can reproduce the adsorption trend very well from low pressure to high pressure.

Regarding the outliers in Figure 2a, they may originate from two aspects: the challenging-to-learn samples and those with labeling errors, which can be considered as Bayesian errors to some extent. However, this tends to have little impact on the ranking results of the model, especially for a large database. Our predictive results also support this claim. Although systemic errors are unavoidable, there are methods to alleviate them. First, we can collect a wider range of pre-training databases,

including databases from different sources (computational or experimental), and obtain more robust representations of MOF structures through pre-training based on massive data, thus improving the generalization of downstream predictions. Secondly, we can conduct a more in-depth analysis of outliers to exclude the influence of outliers. Due to the huge amount of hMOF database (more than 2,400,000 adsorption data points), we believe that the study of hard examples and outliers should also be explored in the future. We added the analysis to make it clearer.

(Page 5) **Figure 2**

(Supporting Information, Page 2) **Table S3**

(Supporting Information, Page 2) **Table S4**

(Supporting Information, Page 3) **Figure S1**

(Supporting Information, Page 3) **Figure S2**

(Supporting Information, Page 4) **Table S5**

(Supporting Information, Page 19, Method) **"Loss function for regression ..."**

(Page 5, paragraph 1) "The analysis also incorporates two other error measures, *i.e.*, MAE and RMSE. The hMOF and CoRE_MAP datasets exhibit low MAE and RMSE values, indicating acceptable outlier errors in both the hMOF and CoRE_MAP databases. However, the CoRE_MOFX dataset shows larger errors, particularly in RMSE. CoRE_MOFX dataset contains Ar and N₂ adsorption amounts at 77K and 87K, which are significantly lower temperatures compared to the other two databases. Consequently, the adsorption values in CoRE_MOFX are generally larger. The adsorption isotherms with the most significant errors are depicted in Figure S2. MOF materials with the greatest error typically have Cu (with the lowest energy parameter of 2.52 Å) as the metal site. This implies that MOF materials with low energy parameter metal sites or extremely large pore sizes are the limiting factors for low-temperature data prediction. Intriguingly, even with the highest errors, Uni-MOF can accurately reproduce the adsorption trend from low to high pressure."

(5) In Fig. 2a, it is observed that while the simulated values keep increasing beyond 9-10 mol/kg, the predicted maximum uptake value remains around 9-10 mol/kg. This suggests that if the developed model was used to predict the gas uptake values in high performing MOFs (with simulated uptake values beyond 14 mol/kg), it would not be able to provide realistic uptake values making it hard to identify the best performing MOFs. This implies that the use of the developed model for a large MOF database may help identify large groups of high and low-performing MOFs but not lead to quantitative differences in terms of gas storage performance for high-performing MOFs as the predicted values would be very similar. Given the excessive number of MOFs generated so far (in the order of hundreds of trillions, <https://doi.org/10.1021/acsami.1c02471>), the use of the developed model for large MOF databases could result in having excessively long lists of structures rather than "shortlists" for which performing computational simulations could be practical. Considering such a case, what strategy could be implemented to differentiate the performance disparities across many high-performing MOFs that could be identified by the developed model?

[Authors' Reply] We thank the referee for the consideration of model performance for high-performance MOFs. After analyzing the data, we found that less than 0.3% of the data points in the hMOF database had adsorption capacity exceeding 10 mol/kg. Due to the large amount of hMOF data, it was difficult for the model to learn the value of high adsorption capacity. In order to demonstrate the robustness of Uni-MOF, we re-optimized the model by increasing the parameter "epoch", that is, increasing the number of training cycles. We agree with the referee's point that to obtain a shortlist through Uni-MOF high-throughput screening, which can be further verified by simulations. Meanwhile, the optimized Uni-MOF still shows the excellent learning ability of high-performance MOF. This part of the training used two GPUs and took around a month, Figure 2a was renewed to demonstrate the results.

(Page 5) **Figure 2a**

(6) On page 4, the authors indicate that "The model with the highest validation set R² (coefficient of determination) was saved as the optimal model, and the final R² was reported using the weights of this model to reasonably avoid over-fitting." Considering this sentence, it is not quite clear how the overfitting was avoided. Could the authors please provide more details on how they ensured the robustness of model and reduced the risk of overfitting?

[Authors' Reply] We thank the referee for asking more details of over-fitting issue. During the model training, the model parameters are continuously optimized and reflected in the results of the validation set. Thus, the optimized results of the model are based on both the training and validation datasets, and ultimately the validation set with the highest R² corresponds to the optimal model. The prediction results of the optimal model in the never-before-seen test set represent the final performance of the model. Therefore, the credibility of the model prediction results is improved by avoiding over-fitting in multiple strategies (dividing the dataset and training the model). To better clarify this confusion, we added the following sentences in the revised manuscript.

(Page 4, paragraph 6) "The model parameters are optimized during the training process and reflected in the results of the

validation set. The optimal model corresponding to the validation set is saved, and the prediction results in the never-before-seen test set represent the final performance (R^2 shown here) of the model, thus reasonably avoiding over-fitting.”

(7) In Fig. 3, many of the predicted values at low pressure are considerably away from the experimental values especially for Mg-dobdc and MOF-5. It is known that simulations may not reproduce experimental values especially when the MOF involves open-metal sites as in Mg-dobdc. However, MOF-5 has closed metal sites and it was shown earlier that simulations could describe the experimental gas uptake values well (<https://doi.org/10.1021/am506793b>). Given the fact that the developed model relies on simulations which can reproduce experimental values for structures like MOF-5, what could be the reason(s) for having highly inaccurate predictions at low pressure for CO₂ and CH₄ uptake in MOF-5?

[Authors’ Reply] We thank the referee for pointing this critical issue out. In this section, we particularly experience the challenges of machine learning engineering when comparing the results with experimental values. While collecting data, we discovered that the adsorption performance of the same gas in the same material can vary, even under identical working conditions. This implies that there is a significant difference in samples and measurements in the experimental values. For instance, in the literature suggested by the referee, the absorption of methane by IRMOF-1 (also known as MOF-5) at a pressure of 400KPa and a temperature of 298K is approximately 2mmol/g (roughly estimated from the figure). At the pressure of 1000KPa, methane absorption is around 4.5mmol/g. However, in the literature we referred to, the methane adsorption capacity is only 0.4mmol/g at 500KPa under the same conditions. This is especially evident in the case of carbon dioxide, with various sources reporting that carbon dioxide absorption in MOF-5 varies from 1.12mmol/g to 2.1mmol/g at a temperature of 298K and a pressure of 1bar. Furthermore, to maintain consistency with the database, the adsorption capacity unit was approximately converted from mmol/g to cm³/g by utilizing the molecular molar mass and the gas volume at Standard Temperature and Pressure (STP), results are listed in Table S6-8.

Due to different experimental methods and objective errors in the experimental values, we did not deliberately select data in this part, but showed all the collected data as far as possible. For example, the black cross points in Figure 3e show the experimental values of different sources under the same working condition. In this section, we promote showcasing the ranking capability of Uni-MOF. Despite significant variations in experimental values, Uni-MOF retains its precision in sorting and can also provide the adsorption quantity as a reference. We added the sentence to clarify this.

(Page 5, paragraph 4) “Nevertheless, we still notice significant deviations between many predicted and experimental values under low pressure, especially in the cases of Mg-dobdc and MOF-5. Simulations may not precisely represent experimental values for MOFs with open metal sites, such as Mg-dobdc. However, MOF-5 has close metal sites, and previous studies have shown that simulations can effectively depict experimental gas adsorption values. Despite these findings, we observed that the gas adsorption performance of MOFs varies even under the same operating conditions (refer to Table S6 - S8). This suggests that different methods and significant objective errors exist in the experimental values. Therefore, we did not purposely select data but instead aimed to provide as comprehensive a representation of the collected data as possible. For example, the black intersections in Figure 3e represent experimental values from various literature sources under identical operating conditions. The results demonstrate that, despite significant variations in experimental values, the Uni-MOF framework maintains a high level of accuracy in material ranking, rendering it suitable for addressing engineering challenges.”

(8) On page 6, it is mentioned that experimental values were used to correct the fitted adsorption isotherms. However, no detail is provided about this correction. Could the authors please elaborate on the details of adsorption isotherm correction?

[Authors’ Reply] We thank the referee for pointing this out. The correction of the adsorption isotherm is actually the result of fitting the adsorption isotherm using both simulated and experimental values. The simulated values alone are used to objectively demonstrate the ability of the model to sort materials at a practical level, and the addition of the experimental values and re-fitting of the adsorption isotherms provides a closer approximation to the actual situation. We made revisions to make it clearer.

(Page 6, paragraph 2) “For example, Figure 3b shows the Langmuir adsorption isotherm obtained by fitting both the predicted and experimental adsorption data. While we use simulated datasets to address data scarcity, we can also properly introduce experimental values to correct adsorption isotherms, which helps a more quantitative prediction of adsorption performance at high-pressure where the gas-gas interaction becomes more significant.”

(9) In Fig. 4g, it is shown that predictions of Kr uptake are more accurate than those of other gas uptakes. It would be great if the authors can provide reasoning for the entire observed trend. For instance, it is interesting that the gas uptake predictions for Xe, which has similar characteristics with Kr, are much less accurate while the prediction accuracy for Ar (another similar inert gas) uptake is somewhere between those of Kr and Xe uptakes. Could the authors please suggest any physical/chemical reason for that?

[Authors’ Reply] We thank the referee for pointing this out and we also notice this. The CoRE_MAP_DB database

contains approximately 100,000 data points, comprising the adsorption capacity of seven gases (methane, carbon dioxide, argon, krypton, xenon, and helium). We randomly divided the data by gas types into three datasets, specifically the training set, the validation set, and the test set (with ratio of 5:1:1). This demonstrates the universal capability of Uni-MOF for cross-gas prediction. Typically, we randomly split the data into three datasets to prevent over-fitting. Nevertheless, transfer learning between various gases poses a significant challenge, particularly when there are a limited number of gas types and data points. In the supplementary information, we divided the data based on gas type into two datasets, namely the training set and the test set (with a 6:1 ratio), to optimize utilization of the limited dataset. The same phenomenon can be seen in Figure S6, even though the prediction accuracy of each gas has improved.

Ar, Kr and Xe are all inert gases, but Kr shows the best prediction performance, while Xe's prediction accuracy is much lower than that of other two inert gases. Since our database is obtained by random sampling, it is found that the distribution of MOF material features in Ar, Kr and Xe databases is very similar, as can be seen from the subfigure in Figure S3. This means that sampling differences in MOF materials have little effect here.

Although the molecular diameter of xenon is larger, which is not conducive to adsorption in small pores, the adsorption limit of xenon is higher than that of argon and Krypton (can be seen in Figure S3 and Figure S4) due to the larger energy parameters (shown in Table S10) when interacting with MOF material atoms. We still analyzed the properties of materials with high xenon adsorption capacity (greater than $400 \text{ cm}^3/\text{g}$), as shown in the yellow part of sub figure in Figure S3c. As expected, materials with high xenon adsorption capacity generally have larger pore size and void fraction, where the LCD is greater than 6.73 \AA , the void fraction is greater than 0.68 and PLD is no less than 4.18 \AA (very close to the kinetic diameter of xenon), thus providing sufficient space for the xenon adsorption process. Although it is difficult to find a universal connection between the model predictions and the physicochemical properties of gases, we can still find that gases with larger energy parameters generally have lower predictive performance (CH_4 , CO_2 and Xe). This is because when the pressure increases, the interaction between gas molecules becomes more and more important, and high energy parameters complicate the situation, thus increasing the difficulty of transfer learning. However, we can still see exciting results, in the same class of inert gases, through the adsorption behavior of argon and xenon, the model can accurately predict the adsorption of moderate krypton gases. This means that transfer learning is feasible in our model, which is one of the reasons we came up with the unified model. We made the following revisions to clarify this.

(Supporting Information, Page 7) Table S9

(Supporting Information, Page 7) Table S10

(Supporting Information, Page 8) Figure S3

(Supporting Information, Page 9) Figure S4

(Supporting Information, Page 10) Figure S5

(Supporting Information, Page 10) Table S11

(Supporting Information, Page 11) Figure S6

(Page 7, paragraph 4) "One can observe that prediction performance varies among gases. For instance, Ar, Kr, and Xe are all noble gases with Kr demonstrating the best performance among them with R^2 of 0.85, while the prediction performance of Xe is inferior ($R^2 = 0.41$). Similar distribution of MOF material features across Ar, Kr, and Xe databases is observed, as illustrated in the corresponding subfigure of Figure S3, indicating that MOF material sampling differences have minimal impact in this case. Despite larger molecular diameter of Xe making it less favorable for adsorption in small pores, its higher adsorption limit (depicted in Figure S4) compared to argon and krypton results from its greater energy parameters ($\epsilon = 167.1 \text{ K}$) when interacting with MOF material atoms. In Figure S3c subfigure, materials with high Xe adsorption capacity (exceeding $400 \text{ cm}^3/\text{g}$) are highlighted in yellow. These materials generally exhibit larger pore sizes and void fractions, possessing an LCD greater than 6.73 \AA , a void fraction exceeding 0.68 , and a PLD of at least 4.18 \AA (close to the kinetic diameter of xenon with 3.96 \AA). Although it is challenging to establish a comprehensive connection between model predictions and the physicochemical properties of gases, it is discernible that gases possessing larger energy parameters typically demonstrate inferior predictive performance, such as CH_4 , CO_2 and Xe. This phenomenon can be attributed to the increased significance of gas molecule interactions with escalating pressure, coupled with the complexity introduced by high energy parameters, ultimately complicating transfer learning. Notwithstanding the challenges, encouraging outcomes can still be observed, as the model precisely predicts the adsorption of moderate krypton, premised on the adsorption behavior of argon and xenon within the same inert gas category."

(10) In Fig.5, considering the cases where the ML models were trained individually for small data sets, the prediction accuracies of some of those cases are higher than those of Uni-MOF which was trained with the entire data set. Could the authors please elaborate on the underlying reasons for this observation?

[Authors' Reply] We thank the referee for this and it is indeed a good question. In this part of the experiment, we divided the original data set into three datasets of train, valid and test with ratio of 8:1:1. Each data set contains a subset of each system, for example, the overall training set contains the training set of each small system. After training the entire data set, we calculate

the predicted performance of each small system individually. Therefore, all the diamond dots in the figure represent the overall predicted performance of Uni-MOF. In fact, it is normal for Uni-MOF to differ in performance in different subsystems, and the distribution of different data sets and the influence of high and low pressure on gas adsorption will be reflected in the predicted results. However, we can see that the results of training a single system individually in the true sense, that is, the cross points in the figure, do not exceed the overall performance of UniMOF. This also means that extensive data sampling can further help the model enhance its generalization ability. We made revisions to make this clearer.

(Page 8, paragraph 2) “In this part, the data sets are divided in a manner where each set comprises the relevant subsystem data. Therefore, all the diamond dots in Figure 5a represent the overall predicted performance of Uni-MOF. Indeed, Uni-MOF demonstrates equally impressive performance on the complete data set as it does on an individual data set, even when the performance of a single data set relies on a more extensive data than its specific prediction task.”

(11) In Fig. 6a, it is shown that the predicted PLD values for CoRE MOFs (especially with large PLDs) could be considerably away from the parity line while those for the hMOFs are very close to the parity line. Could the authors explain why the PLD predictions for synthesized MOFs could be less accurate than those for the hypothetical MOFs?

[Authors’ Reply] We thank the referee for pointing this out. Through the combination of structural building blocks (metal nodes, ligands and functional groups), we explored the design space beyond experimentally known MOF structures, *i.e.*, hypothetical MOF materials (hMOFs). Experimentally synthesized MOF materials (CoRE_MOF) tend to have structural preferences and sparsity in the combinations of building blocks, e.g., zinc, copper and cadmium are the most common metal sites in them. In contrast to CoRE_MOF, hMOF dataset fully samples different structural building blocks to form a comprehensive structural library. As a result, the distribution of data points varies greatly from one database to another. Literature has proved that the experimental MOF (CoRE-2019) is mainly distributed in the small pore region, while the hMOF (Tobacco) is biased towards the large pore region. The same observation can be found in Fig. S7, where Figure S7a and Figure S7b show the differences between CoRE_MOF and hMOF structures using the T-SNE method. Through Fig. S7c-f, it can be seen that the apertures of CoRE_MOF are obviously concentrated in the small aperture region, especially the PLD. hMOF, on the other hand, can still observe the aperture gradient due to its rich structural sampling and data points, even though the apertures are generally small. Thus, for different types of databases, the prediction of structural features performs differently, especially for PLD.

(Supporting Information, Page 15) Figure S7

(Supporting Information, Page 16) Figure S8

(Page 9, paragraph 4) “It can be observed that the prediction performance of hMOF structural parameters is generally superior than that of CoRE_MOF. By employing a combination of structural building blocks (metal nodes, ligands and functional groups), the design space of hypothetical MOF (hMOFs) dataset was fully explored and structural features are more even distributed compared to CoRE_MOF dataset. Experimentally synthesized MOF materials (CoRE_MOF) tend to have structural preferences for metal nodes (e.g., with zinc, copper, and cadmium) and linkers. Previous research has demonstrated that the experimental MOF (CoRE-2019) is primarily concentrated in the small pore region, while the hMOF (Tobacco) is more even distributed with a slight shift of focus towards the large pore region. The same observation can be found in Figure S7, where Figure S7a and b show the differences between CoRE_MOF and hMOF structures using the T-SNE method. Through Figure S7c-f, it can be seen that the apertures of CoRE_MOF are obviously concentrated in the small aperture region, especially the PLD. hMOF, on the other hand, can still observe the aperture gradient due to its rich structural sampling and data points, even though the apertures are generally small. Thus, for different types of databases, the prediction of structural features performs differently, especially for PLD.”

(12) What SBUs were selected for the generation of more than 300,000 MOFs and what were the reasons to select those SBUs?

[Authors’ Reply] We thank the referee to help better clarify this part. The 306,773 MOF structures were generated following the code of ToBaCCo_3.0 as provided in this link. The ToBaCCo program takes as input a topological blueprint, searches compatible node building blocks from a defined set, and constructs all possible structures using the topology in combination with different node building blocks and edge building blocks. In practice, the program generates structures with three folders as input variables, *i.e.*, ‘templates’, ‘nodes’, ‘edges’. To generate as many MOF structures as possible, we use all edges as provided in ‘edges_database’, all nodes as provided in ‘nodes_database’ and all templates as provided in ‘template_database’. For example, 5 templates used for generating 300,000 MOF structures are listed as “nnd.cif”, “tsl.cif”, “bce.cif”, “hfp.cif”, “xfe.cif”. Finally, there are 2678, 47, and 45 cif files used as template, edges, and nodes, respectively.

(Page 12, **Method**) “The ToBaCCo program takes as input a topological blueprint, searches compatible node building blocks from a defined set, and constructs all possible structures using the topology in combination with different node building blocks and edge building blocks. In particular, the program generates structures with three folders as input variables, *i.e.*, ‘templates’, ‘nodes’, ‘edges’. To generate as many MOF structures as possible, we use all edges as provided in ‘edges_database’,

all nodes as provided in 'nodes_database' and all templates as provided in 'template_database'."

(13) Could the authors please discuss the limitations of the developed model? For instance, how safe would it be to get gas uptake predictions with the developed model at the unexplored conditions (physically/chemically different MOFs, temperature/pressure out of the bounds investigated)? In short, do the authors think that the developed model is fully universal for the adsorption of gases investigated in MOFs?

[Authors' Reply] We thank the referee for reminding us of the model universality. In fact, our database contains more than 470,000 materials, including CoRE_MOF and hMOF, with nearly 40 different metal sites, pore sizes ranging from 2 to 40 Å, and other structural features that vary. Uni-MOF has demonstrated precise prediction outcomes for a fairly comprehensive collection of MOF structures. On the other hand, COF materials lack metal sites and possess a more loosely arranged spatial structure. Despite the distinct physical and chemical properties of COF materials, Uni-MOF demonstrates exceptional predictive capabilities for them as well.

Furthermore, the database encompasses a pressure range of 0-10 bar and temperature ranges of 77K-87K and 150-300K, making it suitable for the majority of gas adsorption issues. The prediction outcomes indicate that the overall performance of our model is outstanding. Even though there are systematic errors in the model, such as predicting the adsorption of certain MOFs at lower temperatures, the provided prediction trend remains highly reliable.

Above all, we consider Uni-MOF to be a completely universal framework, since it has had outstanding performance with existing databases. However, the use case of the model is always changing, which is why we propose a universal "framework" rather than an established "model". With Uni-MOF, the continuous updating of databases and even usage scenarios are supportable, which is another meaning of universality. We made the revisions to clarify this.

(Page 5, paragraph 1) "The analysis also incorporates two other error measures, *i.e.*, MAE and RMSE. The hMOF and CoRE_MAP datasets exhibit low MAE and RMSE values, indicating acceptable outlier errors in both the hMOF and CoRE_MAP databases. However, the CoRE_MOFX dataset shows larger errors, particularly in RMSE. CoRE_MOFX dataset contains Ar and N₂ adsorption amounts at 77K and 87K, which are significantly lower temperatures compared to the other two databases. Consequently, the adsorption values in CoRE_MOFX are generally larger. The adsorption isotherms with the most significant errors are depicted in Figure S2. MOF materials with the greatest error typically have Cu (with the lowest energy parameter of 2.52 Å) as the metal site. This implies that MOF materials with low energy parameter metal sites or extremely large pore sizes are the limiting factors for low-temperature data prediction. Intriguingly, even with the highest errors, Uni-MOF can accurately reproduce the adsorption trend from low to high pressure."

(Page 7, paragraph 6) "Furthermore, we investigated the capability of Uni-MOF to predict the gas adsorption behavior in COFs. Around 500 data points of CH₄ adsorption uptake in CoRE COFs at 300K were simulated. Despite the limited size of COF adsorption database, Uni-MOF maintains outstanding predictive performance, achieving an R^2 of 0.76. Additionally, as shown in Figure 4h, Uni-MOF exhibits outstanding ranking capability for the adsorption of diverse materials under various pressures."

(Page 12, paragraph 3) "Our database encompasses a pressure range of 0-10 bar and temperature ranges of 77-87 K and 150-300 K, making it suitable for the majority of gas adsorption issues. Uni-MOF has demonstrated precise prediction outcomes for a fairly comprehensive collection of MOF structures and distinct COF materials. Even under some extreme conditions, the predicted trend remains highly dependable. With Uni-MOF, the continuous updating of databases and even usage scenarios are supportable, which further emphasizes its universality."

There are also a number of minor issues mentioned below that needs to be handled:

(1) On page 4, it is mentioned that the data set was divided into 8:1:1 which very likely refers to ratios of training, validation, testing sets, respectively. However, this is not explicitly mentioned. Could the authors please mention the sizes of those sets in the text to be clearer? Also, in the same sentence the authors mention that the data set was not randomly split. Could the authors elaborate on it more to help readers better understand how exactly the data splitting was made (e.g., stratified sampling)?

[Authors' Reply] We thank the referee for these suggestions. The 8:1:1 division refers to the ratio of training, validation, testing sets, respectively. In order to make it clearer, we revised the following sentence.

(Page 4, paragraph 6) "To prevent data bias and ensure that the test set remained unseen by the model, we divided the data set into three different data sets (train, valid and test) with the ratio of 8:1:1 according to the MOF structure instead of randomly splitting, that is, there is no identical material between the three datasets."

Since different databases contain gas uptake data under different conditions (multiple MOFs, temperatures, pressures, gas molecules), this may result in providing excess information to the computer, which reduces robustness. For example, if the

computer has seen the gas adsorption of a certain material at low pressure in the training or validation set, the high-pressure adsorption for the same scenario in the test set can be easily extrapolated, resulting in overfitting. There is a wide variety of materials whose morphology and internal structure can critically affect gas adsorption. Therefore, when we divide the datasets (training, validation, and test sets), we do not directly use the random division, but through the material division to ensure that the model needs to complete the prediction of new materials in each validation and test, instead of those materials that have already been seen. We also added the following sentences in the revised manuscript.

(Page 4, paragraph 6) “Through the material division to ensure that the model accomplishes the prediction of new materials in each validation and test set, instead of those materials that have already been seen.”

(2) In Fig. 3, it seems lines represent Langmuir fits. Please indicate it in the figure caption for clarity if that is the case.

[Authors’ Reply] We thank the referee for pointing this out. Lines in Fig. 3 represent Langmuir fits indeed, we made revision in caption to clarify this.

(Page 6, Fig. 3 Caption) “Adsorption isotherms based on low-pressure predictions and high-pressure experimental values, each curve represents Langmuir fit.”

(3) In the caption of Fig. 4, it is mentioned that “The adsorption isotherms are obtained only by simulated value.” however the caption also states that “The lighter color dot represents the predicted adsorption uptake of Uni-MOF” which makes the first statement confusing as it is implied that adsorption isotherms were obtained using ML model and simulations. Did the authors mean the adsorption isotherms were simulation-based and not experiment-based? If that is the case, the first statement seems redundant. Could the authors please clarify it?

[Authors’ Reply] We thank the referee for reminding us of these details. “The adsorption isotherms are obtained only by simulated value.” here the adsorption isotherm means the grey curve, which is included in the next topic. We delete the first statement for misleading.

(4) In Fig. 4a-f, do different markers denote different temperatures? If yes, it would be better to add legend(s) mapping marker shapes to temperatures. Also, it is not clear in those subplots how gray lines were plotted? Looking at Fig. 4e, the lines do not go through all markers which suggests that those lines were not drawn to guide the eye.

[Authors’ Reply] We thank the referee for the careful reviewing. In Fig. 4a-f, different markers represent different MOFs, and each figure denotes the same temperature. Fig. 4a-f were redrawn and revisions were made to make it clearer.

(Page 8, Fig. 4) Temperature text was added.

(Page 8, Fig. 4 Caption) “Different markers in each figure denote different MOF adsorbents.”

In Fig. 4, the lighter color dot represents the predicted adsorption uptake from Uni-MOF, the darker one represents the simulated value. The grey curve were obtained by all simulated values, and the lighter dots are here to show the correlation between predicted and simulated values. If we obtained the curve using both ML model and simulations, there is some correlation we can always find. So here we eliminate data bias and prove the robustness of our model in each single cases. We added the sentences to clarify this.

(Page 7, paragraph 2) “Adsorption isotherms fitting from both Uni-MOF predictions and simulated values would artificially reduce visual errors, in order to eliminate data bias, adsorption isotherms in all cases were obtained only by simulated values.”

(Page 8, Fig. 4 Caption) “The adsorption isotherms are obtained only by simulated values to eliminate data bias.”

(5) On page 12, it is mentioned that void fraction, void volume, and surface area were obtained using Zeo++. Please provide probe sizes used for those computations and specify whether these computations rely on the concept of probe-occupiable volume or probe center pore volume as defined in <https://doi.org/10.1021/acs.langmuir.7b01682>

[Authors’ Reply] We thank the referee for the careful reviewing. The probe size we used here is 1.8 Å, which is close to the dynamic diameter of Nitrogen molecules (3.64 Å). The kinetic diameter is related to the mean free path of the molecules in the gas. Therefore, the structural property calculations in this paper can well simulate the Nitrogen adsorption experiments in nanoporous materials.

We thank the referee for the suggestion, the literature cited is excellent work and very helpful. The cited paper presents volume considered for different methods, especially the accessible and nonaccessible probe center pore volume (Ac-PC, NAc-PC), accessible and nonaccessible probe-occupiable pore volume (Ac-PO, NAc-PO). The accessible and nonaccessible volumes sometimes depend on the size of the probe, a topic to which we have responded above. In our work, accessible volume calculated here is defined as the volume available to the center of spherical probe, corresponding to the accessible probe center pore volume. In addition, probe-occupiable volume is also available in Zeo++, since we only want to validate the well-trained model in structural property prediction, the accessible volume is calculated as the representative property. We

added the sentences to clarify this.

(Page 14, High-throughput analysis of large MOF database) “The internal void volume can largely influence the performance of nanoporous materials, precise definitions existing for volume from different methods, especially the accessible and nonaccessible probe center pore volume (Ac-PC, NAc-PC), accessible and nonaccessible probe-occupiable pore volume (Ac-PO, NAc-PO). The concept of accessibility relies on the probe size, the probe of radius of 1.8 Å is used in our work, in order to directly related to the pore volumes from experimentally measured Nitrogen (kinetic diameter of 3.64 Å) isotherms. Accessible volume calculated here is also defined as the volume available to the center of spherical probe, corresponding to the accessible probe center pore volume.”

(6) Not much simulation details were provided for GCMC simulations such as atomic partial charges, forcefields, molecular models etc. To enable reproducibility of the work, please provide computational details as necessary. Also, on page 11, it is stated that 50,000 “steps” were used as initialization “cycles” and additional 50,000 steps were used for production. Since steps and cycles mean two different things in RASPA, could the authors please clarify whether they used 50,000 steps or cycles for initialization and production?

[Authors’ Reply] We thank the referee for pointing this out. We added these contents to provide necessary details about GCMC simulation.

(Page 12, Materials and data generation) “Interactions of gas molecules with individual material atoms are described in terms of Lennard-Jones (12-6) potential, the force field parameters are obtained from universal force field (UFF).”

We thank the referee for careful reviewing. In fact, there are different steps in a Monte Carlo simulation, such as translation, insertion, deletion, rotation, etc. In our work, we used 50,000 cycles for initialization and production, respectively. Each cycle has attempts for different steps. We made the following revisions to clarify this.

(Page 12, Materials and data generation) “In addition, we conducted Grand Canonical Monte Carlo (GCMC) simulations on RASPA software to produce another 10,000+ gas adsorption uptake dataset, with 50,000 initialization cycles and an additional 50,000 cycles employed for adsorption capacity samples.”

(7) On page 8, it is mentioned that Argon accounts for the vast majority of the atmosphere while its concentration in the atmosphere is actually low, which is also stated in the same sentence. Did the authors mean Argon accounts for the majority of the inert gas content in the atmosphere?

[Authors’ Reply] We thank the referee for pointing this out. Like the referee said, here we mean Argon accounts for the majority of noble gases in the atmosphere. We made revisions to make it clearer.

(Page 10, paragraph 2) “Argon accounts for the vast majority of noble gases in the atmosphere (9340 ppm at ambient conditions), has been widely used for insulation and illumination with a commercial value of 3.1 USD/kg.”

(8) In Fig. 6d and Table S10, it would be better to state the type of “volume” (e.g., pore volume, unit cell volume) to avoid any confusion.

[Authors’ Reply] We thank the referee for this suggestion. We made following revisions to state the type of "volume".

(Page 10, Fig. 6 Caption) “**d** pore volume of MOFs in hMOF and CoRE_MOF databases.”

(Supporting Information Page 14) Table S16

(9) On page 11, it is mentioned that pymatgen was used to extract material properties. Could the authors please be more specific on the material properties calculated using pymatgen? If that is a long list of properties, it could be provided in the Supporting Information.

[Authors’ Reply] We thank the referee for pointing this out. We added the sentence to specify the material properties calculated using pymatgen.

(Page 12, Material analysis) “Materials properties including lattice vectors, lattice angles, unit cell volume, atoms and coordinates are extracted using pymatgen. Where atom types and coordinates will be used in the pre-training stage for self-supervised learning strategy.”

(10) In the Author Contributions section, it is mentioned that three authors conducted breakthrough experiments which is confusing as no breakthrough data were presented.

[Authors’ Reply] We thank the referee for careful reviewing. Here we want to point out that J.W., J.L. and Z.G. collected the comprehensive database and developed a breakthrough unified model. We made revisions to avoid misleading.

(Page 17, Author contributions) “J.W., J.L. and Z.G. performed the data collection, developed the breakthrough model and drafted the paper.”

(11) In the captions of Table S7 and S8, Uni-MOFsingle and Uni-MOFmixed notations were explained however they do not appear in the corresponding table. Instead, Uni-MOF_{I&SP} and Uni-MOF_{MAP} exist in the table which were not defined. Please clarify it.

[Authors' Reply] We thank the referee for pointing this out. In the beginning, we gave names to different models, Uni-MOF_{MAP} is the model for Multi-system Adsorption Property predictions, and Uni-MOF_{I&SP} is the one for Intrinsic or Single-system Property predictions. But after consideration, we want to put the perspective in the comprehensive application scenario of Uni-MOF. So we simply use the names of Uni-MOF_{mixed} and Uni-MOF_{single}. Sorry for the typo, we made modifications to Table S13, S14 and S15.

(Supporting Information, Page 12) **Table S13**

(Supporting Information, Page 13) **Table S14**

(Supporting Information, Page 13) **Table S15**

(12) MOFX-DB which is used as one of the data sources mainly involves single-component gas adsorption results as well as some multi-component gas adsorption results. It would be better if the authors could be more specific on whether the gas adsorption data used to train the models were results of single-component, multi-component gas adsorption simulations (or a combination of both).

[Authors' Reply] We thank the referee for this considerate suggestion. In this work, we only use the single-component gas adsorption results to train our model. We made revisions to clarify it.

(Page 14, Data Availability) **“In this work, only single-component gas adsorption data are considered”**

Reviewer: 2

Recommendation: Do not publish.

Comments: In this work Wang and coworkers present a machine learning (ML) workflow and result for prediction of gas adsorption in MOFs. The most impactful part of the work is the Uni-MOF module which is capable of describing MOFs in a way that is amenable for use in other ML workflows and they demonstrate that it learns textural property information for the MOFs. However, there are claims made in the manuscript that I don't believe are substantiated with the presented data and thus I cannot recommend publication. Namely:

[Authors' Reply] We thank the referee for examining this work with critical comments. We explained the innovation and breakthrough of our work, added new results and analysis to further make our work more convincing.

(1) Claims as far as universality of adsorption predictions go are not really tested or demonstrated. Authors make claims of being able to make adsorption predictions at multiple conditions. However, the only data that is presented is some isotherms in some MOFs with at most 100 K difference in temperature of the isotherms. That is not enough to make a claim of universality of adsorption predictions.

[Authors' Reply] We thank the referee for this suggestion. However, we are afraid that we may not fully agree with the reviewer's comments. Our database encompasses temperature ranges of 77K-87K and 150-300K, making it suitable for the majority of gas adsorption issues. The adsorption data for MOF structures that can be collected online are almost used here. Additionally, we conducted further experiments to augment the dataset and verify the model. The prediction outcomes indicate that the overall performance of our model is outstanding. The "universality" of this model is stressed more on the idea, the framework, and its capability to predict the adsorption values at various conditions.

Above all, we consider Uni-MOF to be a completely universal framework, since it has had outstanding performance with existing databases. However, the use case of the model is always changing, which is why we propose a universal "framework" rather than an established "model". With Uni-MOF, the continuous updating of databases and even usage scenarios are supportable, which is another meaning of universality. We claim the "universality" of the model and would like to stress its capability and this can be done as soon as more adsorption data is available. We added the content to clarify this.

(Page 12, paragraph 3) **“Our database encompasses a pressure range of 0-10 bar and temperature ranges of 77-87 K and 150-300 K, making it suitable for the majority of gas adsorption issues. Uni-MOF has demonstrated precise prediction outcomes for a fairly comprehensive collection of MOF structures and distinct COF materials. Even under some extreme conditions, the predicted trend remains highly dependable. With Uni-MOF, the continuous updating of databases and even usage scenarios are supportable, which further emphasizes its universality.”**

(2) Along the lines of universality and using data from one adsorbate to make predictions about others, authors should compare their work against the recommender system from Cory Simon and the alchemical isotherm ML predictions from Gomez-Gualdron. Authors should comment on this and clearly express what are the advantages of the presented approach.

[Authors' Reply] We thank the referee for asking the model comparison. Since the referee did not recommend specific literature, we searched and read the relevant reports.

In <https://doi.org/10.1073/pnas.1613874114>, **Cory M. Simon** and the coworkers proposed a statistical mechanical model for gas adsorption in porous materials. It's an impressive piece of work. In order to verify the exact solution, their model in the grand-canonical ensemble is simulated using the Metropolis Hastings algorithm. Also, empirical formula is used to combine the model with the experimental results. In the case study of MIL-91(Al) demonstrated in their work, the density functional theory calculation was adopted to capture the conformation, and the Rn, Xe, Kr and Ar adsorption isotherms in MIL-91(Al) were depicted. The calculation method of this work is very advanced, which can calculate the adsorption isotherms of different gases, which is undoubtedly an outstanding work in the field of scientific research. Advanced computational methods are essential for scientific research in some cases, but they often require extremely high computational costs and have specific application scenarios, such as "gas adsorption in a porous crystal whose cages share a common ligand that can adopt two distinct rotational conformations" mentioned in the article. **Our goal is to quickly and accurately predict adsorption results in thousands of materials under different working conditions, based on a well-established engineering model.**

In the realm of AI-assisted materials science, data and advanced models are the two most crucial parts. Our idea aligns with the referee's perspective. To propose an engineering model, we collect and compute database based on Monte Carlo simulation that serves the model. Our research findings also demonstrate that our database can effectively correlate with experimental values. However, this does not imply that our model rejects advanced computational methods, which is why we introduce the Uni-MOF "framework". This engineered framework is not only applicable to various prediction tasks but also supports ongoing updates of the database.

In <https://doi.org/10.1021/acs.jctc.9b00940>, **Gomez-Gualdron** and the coworkers came up with a deep learning model to predict the adsorption isotherm in MOFs, is a very cutting-edge work in 2021. The multilayer-perceptron (MLP) model in their

work contains 28 input variables, including 6 textural properties (such as void fraction, pore limiting diameter, density and etc.) and number density of different atoms. Further, in <https://doi.org/10.1063/5.0048736>, they leveraged the developed MLP model to predict the adsorption-based separation property of binary mixtures. The work here is actually divided into two parts, one is to predict the gas adsorption isotherm, and the other is to translate the single component isotherms to mixture loading through pyIAST iast function (<https://doi.org/10.1016/j.cpc.2015.11.016>). As the prediction of gas adsorption isotherms is crucial, we will focus on this part. They used ToBaCCo0-3.0 to build a 51,520-MOF database, and simulate the adsorption of six gases (Argon, Krypton, Xenon, Nitrogen, Methane, Ethane) under around 15 pressures, however, the temperature is a constant. In other words, they built a database of more than 50,000 MOF materials, and trained models on the adsorption properties of different gases through the process of descriptor extraction.

In our work, transformer algorithm is used to directly take the three-dimensional spatial structure of MOF materials as the input of the model, avoiding complicated descriptor extraction and retaining the structural information of materials as much as possible. In terms of database building, we collected and generated a total of nearly 300,000 adsorption data points, including 8 gases, 31 temperatures (150K - 300K), and 41 pressures (1Pa - 3bar). In addition, we also adopted a strategy of 3D structure pre-training, which allows the model to further learn the material configuration by masking the atomic types and coordinates. Accordingly, we collected and generated a total of more than 600,000 nanoporous materials. Unlike the referenced work which relies on the same model for predicting different adsorption isotherms, we have established a unified framework. This single model allows us to consistently predict adsorption values across a range of operating conditions. Therefore, we have not only established a substantial database, but also developed a unified Uni-MOF framework that can directly identify the three-dimensional spatial structure of materials. The pre-training strategy can further improve the model performance, enabling the Uni-MOF framework to predict different gas adsorption amounts in a large number of materials under a wide range of operating conditions.

(3) Another problem with their claim of universality is that they need to train a model for each database separately. If each database of structures requires separate training, then the claims of universality are not very strong.

[Authors' Reply] We thank the referee to asking the universality of our model. In this work, we introduced the **universal framework** to predict the adsorption performance under varied operating conditions. However, we know that in addition to the development of the state-of-art model, data collection is also a bottleneck problem for machine learning in materials science. Based on this, we have collected a wide range of databases containing nearly three million data points, and the reason why we train each of the three datasets separately is that the data come from different sources, and combining them arbitrarily would greatly reduce the performance of the model. The performance of the three datasets enables the validation of our model in all aspects. The generalization of our framework has been demonstrated when the model can have the ability to accurately rank the experimental data after learning a dataset with less than 100,000 simulated data points.

In conclusion, the focus of the article is to propose a unified framework applicable to gas adsorption practical problems, which not only performs well on a variety of datasets, but also supports modular modification so as to cope with different engineering problems. We added the sentence to clarify this.

(Page 12, paragraph 3) **“Our database encompasses a pressure range of 0-10 bar and temperature ranges of 77-87 K and 150-300 K, making it suitable for the majority of gas adsorption issues. Uni-MOF has demonstrated precise prediction outcomes for a fairly comprehensive collection of MOF structures and distinct COF materials. Even under some extreme conditions, the predicted trend remains highly dependable. With Uni-MOF, the continuous updating of databases and even usage scenarios are supportable, which further emphasizes its universality.”**

(4) Further issues with the claim of universality is the fact that they report a prediction accuracy of “above 0.35” for all unknown gases. That is not a good result, especially when compared to others from the literature that I’ve already highlighted.

[Authors' Reply] We thank the referee for the constructive suggestion. The CoRE_MAP_DB database contains approximately 100,000 data points, comprising the adsorption capacity of seven gases (methane, carbon dioxide, argon, krypton, xenon, and helium). We randomly divided the data by gas types into three datasets, specifically the training set, the validation set, and the test set (with ratio of 5:1:1). This demonstrates the universal capability of Uni-MOF for cross-gas prediction. Typically, we randomly split the data into three datasets to prevent over-fitting. Nevertheless, transfer learning between various gases poses a significant challenge, particularly when there are a limited number of gas types and data points. In the supplementary information, we divided the data based on gas type into two datasets, namely the training set and the test set (with a 6:1 ratio), to optimize utilization of the limited dataset.

In the literature suggested by the referee, those models have accurate prediction results for each gas. In fact, this part corresponds to Figure 4 **a-f** and Figure 5 **a-b** in our work. The Uni-MOF model has been proven to have excellent predictive capabilities in both single and multi-component systems. However, transfer learning for different gases in a unified model is rarely studied. Due to the different molecular sizes, masses, and interaction mechanisms with the host materials of the gases, it

is an extremely difficult problem to deduce the adsorption behavior of unknown gases from the known gas adsorption capacity. In the main text, we demonstrate the general predictive ability of Uni-MOF in gas transfer learning using a database with only less than 100,000 data points. This implies that we did not specifically choose the optimal prediction conditions, and it serves as an "engineering detection" in new application domains. We made revisions to clarify this.

(Supporting Information, Page 7) Table S9

(Supporting Information, Page 7) Table S10

(Supporting Information, Page 8) Figure S3

(Supporting Information, Page 9) Figure S4

(Supporting Information, Page 10) Figure S5

(Supporting Information, Page 10) Table S11

(Supporting Information, Page 11) Figure S6

(Page 7, paragraph 3) "To evaluate the predictive capability of Uni-MOF, we randomly divided the CoRE_MAP_DB data points by adsorbate gas into three datasets (train, valid and test dataset with ratio of 5:1:1), then predicted the adsorption capacity for each gas separately."

(Page 7, paragraph 5) "In this study, we showcase the general performance of Uni-MOF for gas transfer learning, employing a database comprising fewer than 100,000 data points. This suggests that the optimal prediction conditions were not specifically selected, functioning as an "engineering detection" in novel application domains. To mitigate over-fitting, the data is typically divided into three sets. However, transfer learning across diverse gases presents a considerable challenge, especially when confronted with a restricted number of gas varieties and data points. As a result, in order to optimize the utilization of the scarce data, the results obtained by dividing the data into two datasets based on the gas types are also shown in Figure S6. Results revealed an improvement in migration learning accuracy for each gas to some extent."

Here we also analyzed the results. Ar, Kr and Xe are all inert gases, but Kr shows the best prediction performance, while Xe's prediction accuracy is much lower than that of other two inert gases. Since our database is obtained by random sampling, it is found that the distribution of MOF material features in Ar, Kr and Xe databases is very similar, as can be seen from the subfigure in Figure S3. This means that sampling differences in MOF materials have little effect here.

Although the molecular diameter of xenon is larger, which is not conducive to adsorption in small pores, the adsorption limit of xenon is higher than that of argon and Krypton (can be seen in Figure S3 and Figure S4) due to the larger energy parameters (shown in Table S10) when interacting with MOF material atoms. We still analyzed the properties of materials with high xenon adsorption capacity (greater than $400 \text{ cm}^3/\text{g}$), as shown in the yellow part of sub figure in Figure S3c. As expected, materials with high xenon adsorption capacity generally have larger pore size and void fraction, where the LCD is greater than 6.73 \AA , the void fraction is greater than 0.68 \AA and PLD is no less than 4.18 \AA (very close to the kinetic diameter of xenon), thus providing sufficient space for the xenon adsorption process. Although it is difficult to find a universal connection between the model predictions and the physicochemical properties of gases, we can still find that gases with larger energy parameters generally have lower predictive performance (CH_4 , CO_2 and Xe). This is because when the pressure increases, the interaction between gas molecules becomes more and more important, and high energy parameters complicate the situation, thus increasing the difficulty of transfer learning. However, we can still see exciting results, in the same class of inert gases, through the adsorption behavior of argon and xenon, the model can accurately predict the adsorption of moderate krypton gases. This means that transfer learning is feasible in our model, which is one of the reasons we came up with the unified model. We made the following revisions to clarify this.

(Page 7, paragraph 4) "One can observe that prediction performance varies among gases. For instance, Ar, Kr, and Xe are all noble gases with Kr demonstrating the best performance among them with R^2 of 0.85, while the prediction performance of Xe is inferior ($R^2 = 0.41$). Similar distribution of MOF material features across Ar, Kr, and Xe databases is observed, as illustrated in the corresponding subfigure of Figure S3, indicating that MOF material sampling differences have minimal impact in this case. Despite larger molecular diameter of Xe making it less favorable for adsorption in small pores, its higher adsorption limit (depicted in Figure S4) compared to argon and krypton results from its greater energy parameters ($\epsilon = 167.1 \text{ K}$, details shown in Table 11) when interacting with MOF material atoms. In Figure S3c subfigure, materials with high Xe adsorption capacity (exceeding $400 \text{ cm}^3/\text{g}$) are highlighted in yellow. These materials generally exhibit larger pore sizes and void fractions, possessing an LCD greater than 6.73 \AA , a void fraction exceeding 0.68 , and a PLD of at least 4.18 \AA (close to the kinetic diameter of xenon with 3.96 \AA). Although it is challenging to establish a comprehensive connection between model predictions and the physicochemical properties of gases, it is discernible that gases possessing larger energy parameters typically demonstrate inferior predictive performance, such as CH_4 , CO_2 and Xe. This phenomenon can be attributed to the increased significance of gas molecule interactions with escalating pressure, coupled with the complexity introduced by high energy parameters, ultimately complicating transfer learning. Notwithstanding the challenges, encouraging outcomes can still be observed, as the model precisely predicts the adsorption of moderate krypton, premised on the adsorption behavior of argon and xenon within the same inert gas category."

(Page 7, paragraph 5) “In this study, we showcase the general performance of Uni-MOF for gas transfer learning, employing a database comprising fewer than 100,000 data points. This suggests that the optimal prediction conditions were not specifically selected, functioning as an “engineering detection” in novel application domains. To mitigate over-fitting, the data is typically divided into three sets. However, transfer learning across diverse gases presents a considerable challenge, especially when confronted with a restricted number of gas varieties and data points. As a result, in order to optimize the utilization of the scarce data, the results obtained by dividing the data into two datasets based on the gas types are also shown in Figure S6. Results revealed an improvement in migration learning accuracy for each gas to some extent.”

(5) Claims are made about predicting high pressure loading from low pressure loading. While it may be true, it is hardly an impressive result. Simple estimations of pore volume and gas density can provide accurate estimates of saturation loading in porous materials, which combined with Henry’s constant or low pressure data can be used to fit a Langmuir isotherm and obtain similar results.

[Authors’ Reply] We agree with the referee that the traditional method combined with saturation loading and Henry’s constant is feasible to estimate the high pressure loading from low pressure one. Most of the currently proven theories are based on empirical assumptions, and in the authors’ view, the application of machine learning in the field of materials is not only about doing previously unattainable tasks, but also about learning as much as possible about natural laws beyond empirical assumptions. Traditional estimation methods require the calculation of the material structure as well as the low-pressure adsorption data to give the predicted value for a single sample. Our model, on the other hand, requires only the three-dimensional spatial structure of the material, implying material identification accuracy at the atomic level. Sufficient labeling data makes it possible for the model to learn information beyond current human cognition, rather than just limiting itself to existing theories. In addition, the present work is a practice of material integration modeling for gas adsorption problems, and the realization of a truly unified model may be the way forward after the model becomes familiar with a large number of materials. Therefore, we think the engineering practice of machine learning is very crucial. We added the following sentence to clarify this.

(Page 2, paragraph 5) “Uni-MOF framework achieves material recognition accuracy at the atomic level, while the integrated model makes Uni-MOF more applicable to engineering problems. Undoubtedly, accomplishing truly unified models is the future direction of the materials field, rather than just focusing on specialized areas. Uni-MOF is a pioneering practice of Machine Learning in gas adsorption.”

(6) Authors mention that this is a “gas adsorption detector” for MOF materials. I am not sure I understand exactly what they mean by gas detector in this context. This should be clarified.

[Authors’ Reply] We thank the referee for asking more clarification of “gas adsorption detector”. We rewrote the sentence to make it clearer.

(Page 2, paragraph 4) “Uni-MOF, as a comprehensive “gas adsorption detector” for MOF materials, requires only the crystallographic information file (CIF) of the MOF, along with the associated gas, temperature, and pressure parameters, to predict the gas adsorption properties of nanoporous materials over a wide range of operating conditions.”

(7) Authors report R2 values for the various models. Figure 2, panel a seems to have some systematic errors at the highest part of the graph where there seems to be a horizontal line. Authors should expand on this and explain that in detail. Further, this brings up the point that perhaps other error metrics like absolute error, mean absolute error, mean relative error, etc. should also be discussed.

[Authors’ Reply] We thank the referee for the consideration of model performance for high-performance MOFs. After analyzing the data, we found that less than 0.3% of the data points in the hMOF database had adsorption capacity exceeding 10 mol/kg. Due to the large amount of hMOF data, it was difficult for the model to learn the value of high adsorption capacity. In fact, a common practice for such database is to remove outliers. However, in order to demonstrate the robustness of Uni-MOF, we re-optimized the model by increasing the parameter “epoch”, that is, increasing the number of training cycles. This part of the training used two GPUs and took around a month, Figure 2a was renewed to demonstrate the results.

(Page 5) **Figure 2a**

In addition, two other error metrics, mean absolute error (MAE) and root mean square error (RMSE), were added to the discussion of results. The hmof dataset demonstrates superior performance with the highest R^2 value, as well as remarkably low MAE and RMSE. Despite the CoRE_MAP dataset exhibiting a lower R^2 , as a result of extensive sampling and limited data volume, it also presents low MAE and RMSE values. This indicates that the error of outliers are acceptable in the hmof and CoRE_MAP datasets. However, for the CoRE_MOFX dataset, we can see some points deviating from parity line. CoRE_MOFX contains the adsorption amounts of Ar and N₂ at 77K and 87K, which are much lower temperatures than the other two datasets, making the adsorption values in CoRE_MOFX generally large. Here we describe the adsorption isotherms with the greatest errors in Figure S2. Interestingly, even with the largest errors, the predicted adsorption isotherm trends still

match the original data points. At the same time, MOF materials with the greatest error generally have Cu (with the lowest energy parameter) as the metal site. This indicates that MOF materials with low energy parameter metal sites or very large pore size are the bottleneck for prediction at low temperature data. But even so, Uni-MOF can reproduce the adsorption trend very well from low pressure to high pressure. We added the contents to explain this.

(Page 5) **Figure 2**

(Supporting Information, Page 2) **Table S3**

(Supporting Information, Page 2) **Table S4**

(Supporting Information, Page 3) **Figure S1**

(Supporting Information, Page 3) **Figure S2**

(Supporting Information, Page 4) **Table S5**

(Supporting Information, Page 19, Method) **"Loss function for regression ..."**

(Page 5, paragraph 1) "The analysis also incorporates two other error measures, *i.e.*, MAE and RMSE. The hMOF and CoRE_MAP datasets exhibit low MAE and RMSE values, indicating acceptable outlier errors in both the hMOF and CoRE_MAP databases. However, the CoRE_MOFX dataset shows larger errors, particularly in RMSE. CoRE_MOFX dataset contains Ar and N₂ adsorption amounts at 77K and 87K, which are significantly lower temperatures compared to the other two databases. Consequently, the adsorption values in CoRE_MOFX are generally larger. The adsorption isotherms with the most significant errors are depicted in Figure S2. MOF materials with the greatest error typically have Cu (with the lowest energy parameter of 2.52 Å) as the metal site. This implies that MOF materials with low energy parameter metal sites or extremely large pore sizes are the limiting factors for low-temperature data prediction. Intriguingly, even with the highest errors, Uni-MOF can accurately reproduce the adsorption trend from low to high pressure."

Reviewer: 3

Recommendation: Publish after minor revisions

Comments: Wang and et. al. report on the development of the Uni-MOF. This graph neural network can accurately predict the gas adsorption of MOFs and COFs with various adsorbates at various experimental conditions, while only relying on the crystal structure for featurization. The authors demonstrate the utility of their models for finding high-performance adsorbents at high pressure, while only using low-pressure data in their training set. Because the model is trained for predictions with a wide variety of adsorbates, I believe that this work could have a significant impact on the gas separation field and can be employed in high-throughput virtual screening workflows. The authors were thorough in their analysis, not only demonstrating that pre-training their models improves model accuracy but also highlighting that pre-trained features also uncover structural features of these materials, such as surface area. Furthermore, the authors demonstrate that the model doesn't simply rely on learning from adsorbate and experimental conditions, showing that the model accuracy doesn't degrade when focusing on a single adsorbate and a constant experimental condition, highlighting that MOF/COF learned features play an integral part in model's predictions. With that all being said, I have two comments I would like the authors to address.

[Authors' Reply] We thank the referee for reviewing our manuscript with critical comments and suggestions. We added new results to further improve the quality of our work.

In the section on cross-system forecasting, I believe that the authors are overstating the performance of their models when employing group splits using adsorbate identity. While performance with Kr is impressive, in other cases performance is mediocre and I'm not sure if any of the adsorbate features are playing any role in the predictions here, or if the model is instead predicting based on the general capabilities of different adsorbents and experimental conditions.

[Authors' Reply] We thank the referee for this suggestion. The CoRE_MAP_DB database contains approximately 100,000 data points, comprising the adsorption capacity of seven gases (methane, carbon dioxide, argon, krypton, xenon, and helium). We randomly divided the data by gas types into three datasets, specifically the training set, the validation set, and the test set (with ratio of 5:1:1). This demonstrates the universal capability of Uni-MOF for cross-gas prediction. Typically, we randomly split the data into three datasets to prevent over-fitting. Nevertheless, transfer learning between various gases poses a significant challenge, particularly when there are a limited number of gas types and data points. In the supplementary information, we divided the data based on gas type into two datasets, namely the training set and the test set (with a 6:1 ratio), to optimize utilization of the limited dataset. The same phenomenon can be seen in Figure S6, even though the prediction accuracy of each gas has improved.

Ar, Kr and Xe are all inert gases, but Kr shows the best prediction performance, while Xe's prediction accuracy is much lower than that of other two inert gases. Since our database is obtained by random sampling, it is found that the distribution of MOF material features in Ar, Kr and Xe databases is very similar, as can be seen from the subfigure in Figure S3. This means that sampling differences in MOF materials have little effect here.

Although the molecular diameter of xenon is larger, which is not conducive to adsorption in small pores, the adsorption limit of xenon is higher than that of argon and Krypton (can be seen in Figure S3 and Figure S4) due to the larger energy parameters (shown in Table S10) when interacting with MOF material atoms. We still analyzed the properties of materials with high xenon adsorption capacity (greater than $400 \text{ cm}^3/\text{g}$), as shown in the yellow part of sub figure in Figure S3c. As expected, materials with high xenon adsorption capacity generally have larger pore size and void fraction, where the LCD is greater than 6.73 \AA , the void fraction is greater than 0.68 \AA and PLD is no less than 4.18 \AA (very close to the kinetic diameter of xenon), thus providing sufficient space for the xenon adsorption process. Although it is difficult to find a universal connection between the model predictions and the physicochemical properties of gases, we can still find that gases with larger energy parameters generally have lower predictive performance (CH_4 , CO_2 and Xe). This is because when the pressure increases, the interaction between gas molecules becomes more and more important, and high energy parameters complicate the situation, thus increasing the difficulty of transfer learning. However, we can still see exciting results, in the same class of inert gases, through the adsorption behavior of argon and xenon, the model can accurately predict the adsorption of moderate krypton gases. This means that transfer learning is feasible in our model, which is one of the reasons we came up with the unified model. We made the following revisions to clarify this.

(Supporting Information, Page 7) **Table S9**

(Supporting Information, Page 7) **Table S10**

(Supporting Information, Page 8) **Figure S3**

(Supporting Information, Page 9) **Figure S4**

(Supporting Information, Page 10) **Figure S5**

(Supporting Information, Page 10) **Table S11**

(Supporting Information, Page 11) **Figure S6**

(Page 7, paragraph 4) “One can observe that prediction performance varies among gases. For instance, Ar, Kr, and Xe are all noble gases with Kr demonstrating the best performance among them with R^2 of 0.85, while the prediction performance of Xe is inferior ($R^2 = 0.41$). Similar distribution of MOF material features across Ar, Kr, and Xe databases is observed, as illustrated in the corresponding subfigure of Figure S3, indicating that MOF material sampling differences have minimal impact in this case. Despite larger molecular diameter of Xe making it less favorable for adsorption in small pores, its higher adsorption limit (depicted in Figure S4) compared to argon and krypton results from its greater energy parameters ($\epsilon = 167.1$ K, details shown in Table 11) when interacting with MOF material atoms. In Figure S3c subfigure, materials with high Xe adsorption capacity (exceeding $400 \text{ cm}^3/\text{g}$) are highlighted in yellow. These materials generally exhibit larger pore sizes and void fractions, possessing an LCD greater than 6.73 \AA , a void fraction exceeding 0.68, and a PLD of at least 4.18 \AA (close to the kinetic diameter of xenon with 3.96 \AA). Although it is challenging to establish a comprehensive connection between model predictions and the physicochemical properties of gases, it is discernible that gases possessing larger energy parameters typically demonstrate inferior predictive performance, such as CH_4 , CO_2 and Xe. This phenomenon can be attributed to the increased significance of gas molecule interactions with escalating pressure, coupled with the complexity introduced by high energy parameters, ultimately complicating transfer learning. Notwithstanding the challenges, encouraging outcomes can still be observed, as the model precisely predicts the adsorption of moderate krypton, premised on the adsorption behavior of argon and xenon within the same inert gas category.”

(Page 7, paragraph 5) “In this study, we showcase the general performance of Uni-MOF for gas transfer learning, employing a database comprising fewer than 100,000 data points. This suggests that the optimal prediction conditions were not specifically selected, functioning as an “engineering detection” in novel application domains. To mitigate over-fitting, the data is typically divided into three sets. However, transfer learning across diverse gases presents a considerable challenge, especially when confronted with a restricted number of gas varieties and data points. As a result, in order to optimize the utilization of the scarce data, the results obtained by dividing the data into two datasets based on the gas types are also shown in Figure S6. Results revealed an improvement in migration learning accuracy for each gas to some extent.”

Finally, I think the quality of the work would be significantly enhanced if the authors could employ some feature importance analysis to get more chemical insights into factors influencing gas adsorption capacities, or differing adsorption capacities between adsorbates. The biggest drawback of GNNs is their black-box nature, and significant progress has been made in recent years toward more explainable AI, and some of these approaches could be implemented in this work to make this work more interesting to experimentalists.

[Authors’ Reply] We thank the referee for this constructive suggestion. In fact, since our model can directly identify the three-dimensional spatial structure of MOF materials, Uni-MOF has no intuitive input features in contrast to the way descriptors are extracted. However, multi-head attention mechanism in Transformer can still learn the atomic interactions within the material structure. Since 64 Heads were used in our model, we present the atomic interaction landscapes of hmof-5004238 material within different Heads in Supporting Information. We also added the contents for explainable machine learning.

(Page 11) Figure 7g, h, i

(Supporting Information, Page 19) Figure S13

(Page 11, paragraph 4) “Furthermore, the multi-head attention mechanism of the Transformer can learn the interactions within the material structure. With the 64-Heads attention algorithm, the atomic interaction landscape of hmof-5004238 in two different heads are represented in Figure 7h, i. As shown in Figure 7h, strong interactions can be observed between the metal sites (Zn), Zn and O atom, and also O atoms. Figure 7i also depicts the interaction between the linear carbon chains (C7-C15). In addition, there is a noticeable correlation between the O atom and the Zn atoms (Zn0, Zn2, Zn4, Zn6). The structural illustration in Figure 7G further confirms this result, as the four atoms are chemically linked. Beyond this, the various chemical landscape of hmof-5004238 in different heads are depicted in Figure S13. In this manner, Uni-MOF identifies tremendous materials and provide reliable predictions for diverse properties.”

REVIEWER COMMENTS

Reviewer #1 (Remarks to the Author):

The authors have significantly improved the manuscript, however there are remaining issues, as noted below, that should be addressed before the manuscript can be published.

1. On page 5, it is discussed that the materials with low energy parameters or extremely large pore sizes could constitute some of the outliers in Figure 2. It would be interesting to indicate what percentage of these outliers involve open-metal sites as the force-field used in the simulations is known to have deficiencies in accurately describing the open-metal site-adsorbate interactions and the simulated values may not serve as accurate reference values.
2. In Table S5, ja5111317_ja5111317_si_003_clean is given as an example of outliers. In CoRE MOF database, it appears to be one of the structures with structural disorder. Did the authors involve the foregoing structure in the training/test set? If yes, did the authors resolve the structural disorder before using it for ML model development? Also, could the authors indicate in the manuscript the number of structures with disorders, if any, used during the development of Uni-MOF? If there were many structures with disorders, it would be better to have a model developed excluding those problematic structures which could improve the prediction accuracy of the model.

There are a few minor issues mentioned below that should be also handled.

1. Please include the units of MAE and RMSE throughout the manuscript including the figures.
2. The unit of energy parameter of Cu atom is incorrect on page 5.
3. Please ensure that the numbers of significant figures in the tables of Supporting Information are consistent.
4. Please provide more details on the GCMC simulations as the specific molecular models used in the simulations (e.g., TraPPE, EPM2) were not mentioned. What method was used to assign partial charges to framework atoms? Was tail-correction used for LJ-potential?
5. The phrase "gas adsorption detector" should be revised in the manuscript to avoid confusion, as the developed model is presented as a gas adsorption predictor rather than a low concentration gas detector/sensor.

Reviewer #2 (Remarks to the Author):

All comments are addressed, paper can be published.

Reply to Comments

Reviewer: 1

Comments: The authors have significantly improved the manuscript, however there are remaining issues, as noted below, that should be addressed before the manuscript can be published.

[Authors' Reply] We really appreciate the referee for the valuable feedback and comments. We've revised the manuscript accordingly as following.

(1) On page 5, it is discussed that the materials with low energy parameters or extremely large pore sizes could constitute some of the outliers in Figure 2. It would be interesting to indicate what percentage of these outliers involve open-metal sites as the force-field used in the simulations is known to have deficiencies in accurately describing the open-metal site-adsorbate interactions and the simulated values may not serve as accurate reference values.

[Authors' Reply] We thank the referee for reminding us to discuss the open-metal sites in outliers. Here, we screened those materials with the largest absolute prediction errors (top 100) from the CoRE_MOFX_DB database. The "OpenMetalDetector" package (https://github.com/emmhald/open_metal_detector) published in DOI: 10.1021/acs.jced.9b00835 was used to calculate the open metal sites in different MOF materials. According to the analysis, outliers with metal open sites accounted for more than 70% of the total. We listed outlier information in Supporting Information, including whether there is an open metal site and whether it is a disordered material. In order to better clarify this, we also made the following revisions.

(Supporting Information, Page 2) Table S3

(Supporting Information, Page 3) Table S6

(Supporting Information, Page 3) Table S7

(Page 5, paragraph 2) “The force field used in this work does not account the effect of open-metal sites. For the top 10 outliers in the CoRE_MOFX_DB database (shown in Table S6), 80% of them have open-metal sites. This suggests that the significant deviations between simulations and Uni-MOF predictions could be due to the missing interaction between open-metal sites and adsorbate considered in the simulation.”

(Page 13, Material analysis) “The Python package - OpenMetalDetector was used to analyze collections of Metal Organic Frameworks for open metal sites.”

(2) In Table S5, ja5111317_ja5111317_si_003_clean is given as an example of outliers. In CoRE MOF database, it appears to be one of the structures with structural disorder. Did the authors involve the foregoing structure in the training/test set? If yes, did the authors resolve the structural disorder before using it for ML model development? Also, could the authors indicate in the manuscript the number of structures with disorders, if any, used during the development of Uni-MOF? If there were many structures with disorders, it would be better to have a model developed excluding those problematic structures which could improve the prediction accuracy of the model.

[Authors' Reply] We thank the referee for pointing out the disordered structures and helping us better discuss the model performance in depth. For datasets used in fine-tuning tasks, only CoRE_MOFX_DB, as the collected experimental structural database, contains 1,800+ disordered materials out of the total of 12,000+ MOFs. During the training process, the database is randomly divided into three datasets by the type of materials, so the training/test sets inevitably contain disordered materials. Due to the disordered internal structure and limited data on the disordered structures, it is more difficult to accurately predict the adsorption value for those materials, such as outlier "ja5111317_ja5111317_si_003_clean" in Table S3. However, we noticed the influence of disordered materials on model performance and developed/fine-tuned models purely based on ordered structures, which is the modeled fine-tuned with CoRE_MAP_DB dataset. The CoRE_MAP_DB database generated in this work has excluded disordered materials. We can see that even with the fewest data points and more operating conditions, the individual errors of the CoRE_MAP_DB database are much smaller than those of CoRE_MOFX_DB (MAE and RMSE shown in Table S4 and S5 can prove this). This also indicates that material cleaning can significantly improve the model's performance, and would be a suggested model to use for performance prediction of nanoporous materials. We made the following revisions to better clarify this.

(Page 5, paragraph 2) “The analysis also incorporates two other error metrics, *i.e.*, Mean Absolute Error (MAE) and Root Mean Square Error (RMSE).”

(Page 5, paragraph 2) “Additionally, CoRE_MOFX_DB, as the collected experimental structural database, contains 1,800+ disordered materials out of 12,000+ MOFs. Disordered MOFs were found to adversely affect the model performance, as the unrealistic internal structures and limited material samples lead to the challenging prediction of gas adsorption, as demonstrated by the outlier "ja5111317_ja5111317_si_003_clean" and "LELDOX_clean" in Table S3. Therefore, disordered materials were excluded from the generated CoRE_MAP_DB database, which results in reduced errors and is more suggested to use for prediction of gas adsorption in nanoporous materials.”

There are a few minor issues mentioned below that should be also handled.

(1) Please include the units of MAE and RMSE throughout the manuscript including the figures.

[Authors' Reply] We thank the referee for reminding us of the error units. Numerical results are listed in Table S4 and Table S5. Since different databases own different units of RMSE and MSE, we delete Figure S1 to avoid misleading comparisons of RMSE and MSE for varied databases.

(Page 5) **Figure 2**

(Page 5, paragraph 3) “As for the downstream tasks, the predictive performance (R^2 , RMSE, and MAE) of Uni-MOF for all three databases surpassed that of Uni-MOF with only MOF pre-training, as illustrated in Table S4 - S5.”

(Supporting Information, Page 3) **Table S4**

(Supporting Information, Page 3) **Table S5**

(Supporting Information, Page 10) **Table S14**

(2) The unit of energy parameter of Cu atom is incorrect on page 5.

[Authors' Reply] We thank the referee for the careful reviewing. We made revisions to correct the unit of energy parameter.

(Page 5, paragraph 2) “MOF materials with the largest error typically have Cu (with the lowest energy parameter of 2.52 K) as the metal site.”

(3) Please ensure that the numbers of significant figures in the tables of Supporting Information are consistent.

[Authors' Reply] We thank the referee for reminding us of these details. Due to the large numerical difference between different parameters, we try to keep the same significant digits for parameters with similar meanings. In addition, even the same parameters can have significantly different values, so we use the same significant number of digits instead of the same decimal number. Therefore, We made the following revisions.

(Supporting Information, Page 1) **Table S1**

(Supporting Information, Page 2) **Table S2**

(Supporting Information, Page 2) **Table S3**

(Supporting Information, Page 3) **Table S4**

(Supporting Information, Page 3) **Table S5**

(Supporting Information, Page 4) **Table S8**

(Supporting Information, Page 5) **Table S9**

(Supporting Information, Page 6) **Table S10**

(Supporting Information, Page 6) **Table S12**

(Supporting Information, Page 10) **Table S14**

(Supporting Information, Page 11) **Table S15**

(Supporting Information, Page 12) **Table S16**

(Supporting Information, Page 12) **Table S17**

(4) Please provide more details on the GCMC simulations as the specific molecular models used in the simulations (e.g., TraPPE, EPM2) were not mentioned. What method was used to assign partial charges to framework atoms? Was tail-correction used for LJ-potential?

[Authors' Reply] We thank the referee for the careful reviewing. In our work, the molecular model TraPPE (Transferable Potentials for Phase Equilibria) was used for gas molecules. In the light of calculation efficiency and consistency, the partial charges of the framework atoms are not considered. In addition, most automated partial charged assignment methods (such as density-derived electrostatic and chemical charge method, DDEC, and approximate charge equilibration method, Qeq) does not yield consistent adsorption results (DOI: 10.1039/C9ME00163H) for large-scale screening, which hinders the direct comparison between experimental results and simulation/prediction results. The force field parameters of framework atoms are adapted from the UFF (Universal Force Field). The cutoff radius is set to 12.9 Å with the consideration of tail-correction. We made following revisions to provide more details.

(Page 12, Materials and data generation) “Interactions between gas molecules and atoms in adsorbent materials were described in terms of Lennard-Jones (12-6) potential, and the cutoff radius is set to 12.9 Å with the tail-correction. The force field parameters of framework atoms were described by UFF (Universal Force Field), and the molecular model TraPPE (Transferable Potentials for Phase Equilibria) was used for gas molecules. In our work, the partial charge of the framework atoms is not considered because of computational cost and significant deviations observed in adsorption results by using different partial charges assignment methods (DOI: 10.1039/C9ME00163H).”

(5) The phrase “gas adsorption detector” should be revised in the manuscript to avoid confusion, as the developed model is presented as a gas adsorption predictor rather than a low concentration gas detector/sensor.

[Authors’ Reply] We thank the referee for the constructive suggestion. In order to avoid confusion, we made the following revisions.

(Page 1, Title) “Uni-MOF: a Comprehensive Transformer-based Approach for High-Accuracy Gas Adsorption Predictions in Metal-Organic Frameworks”

(Page 1, Abstract) "gas adsorption estimator”

(Page 2, paragraph 5) "gas adsorption estimator”

(Page 3) Figure 1

(Page 4, paragraph 4) "The self-supervised learning strategy and abundant databases ensure that Uni-MOF can foretell gas adsorption properties of nanoporous material in a wide range of operating parameters, thereby rendering it a proficient "gas adsorption estimator" for MOF materials.”

(Page 4, Subtitle) "Gas Adsorption Estimator”

(Page 12, paragraph 5) "In summary, Uni-MOF framework serves as a versatile predictive platform for MOF materials, functioning as a "gas adsorption estimator" for MOFs, as it exhibits high precision in predicting gas adsorption under diverse operating conditions and has broad applications in the field of material science.”

(Supporting Information, Page 1, Title) “Uni-MOF: a Comprehensive Transformer-based Approach for High-Accuracy Gas Adsorption Predictions in Metal-Organic Frameworks”

(Supporting Information, Page 1, Subtitle) "Gas Adsorption Estimator”

Reviewer: 2

Comments: All comments are addressed, paper can be published.

[Authors' Reply] We thank the referee for the diligent review and for the approval for publication.

REVIEWERS' COMMENTS

Reviewer #1 (Remarks to the Author):

The manuscript has been improved substantially. As a final edit, please describe and cite the models used for noble gases. Once those details are added, the manuscript can be published.

Part of the instructions to set up the codes requires root permissions. Since I do not have root permissions, I was not able to install NVIDIA Container Toolkit and run the code.

Reply to Comments

Reviewer: 1 (Remarks to the Author)

(1) The manuscript has been improved substantially. As a final edit, please describe and cite the models used for noble gases. Once those details are added, the manuscript can be published.

[Authors' Reply] We really appreciate the referee for the valuable feedback and comments. We've cited the model and revised the manuscript accordingly as following.

(Page 14, paragraph 1) "The force field parameters of noble gas molecules (i.e., argon, krypton and xenon) were estimated from the principle of corresponding states for the second virial coefficients (listed in Supplementary Table 19)."

Reviewer: 1 (Remarks on code availability)

(2) Part of the instructions to set up the codes requires root permissions. Since I do not have root permissions, I was not able to install NVIDIA Container Toolkit and run the code.

[Authors' Reply] We thank the referee for the careful reviewing. The instruction to setup the code requirement permission is due to the Nvidia Container Toolkit installation. The NVIDIA Container Toolkit allows us to run GPU accelerated programs. From the Nvidia official document (<https://docs.nvidia.com/datacenter/cloud-native/container-toolkit/latest/install-guide.html>), we can find that several reason why root access is required:

(1) NVIDIA Container Toolkit installation: The NVIDIA Container Toolkit involves configuring system services and kernel modules, which require root access to modify.

(2) NVIDIA-Docker Configuration: The Docker daemon configuration must be updated to include the 'nvidia-container-runtime' as a default runtime, which requires writing to system directories that are typically not writable by non-root users.

(3) Kernel Modules: Loading the necessary NVIDIA kernel modules (such as 'nvidia.ko', 'nvidia-uvmm.ko', etc.) often requires root access.

If you want to use our software without the NVIDIA root permissions, there are two solutions:

(1) You can use cloud platforms like CoLab and Borihum.

(2) The software supports the CPU version. For details, see Uni-Core of the CPU version. (<https://github.com/dptech-corp/Uni-Core>)

We've added the information above in the installation Readme file of Uni-MOF repository (<https://github.com/dptech-corp/Uni-MOF>).